# The Potential Use of Vitamin D3 and Phytochemicals for Their Anti-Ageing Effects

**DOI:** 10.3390/ijms25042125

**Published:** 2024-02-09

**Authors:** Kazuki Santa, Yoshio Kumazawa, Kenji Watanabe, Isao Nagaoka

**Affiliations:** 1Department of Biotechnology, Tokyo College of Biotechnology, Ota-ku, Tokyo 114-0032, Japan; kazuki_santa@hotmail.com; 2Vino Science Japan Inc., Kawasaki 210-0855, Kanagawa, Japan; 3Department of Biochemistry and Systems Biomedicine, Graduate School of Medicine, Juntendo University, Bunkyo-ku, Tokyo 113-8421, Japan; 4Center for Kampo Medicine, Keio University, Shinjuku-ku, Tokyo 160-8582, Japan; 5Yokohama University of Pharmacy, Yokohama 245-0066, Kanagawa, Japan; 6Faculty of Medical Science, Juntendo University, Urayasu 279-0013, Chiba, Japan

**Keywords:** vitamin D, phytochemicals, anti-ageing, ME-BYO, chronic inflammation, cytokine storms

## Abstract

Unlike other vitamins, vitamin D3 is synthesised in skin cells in the body. Vitamin D3 has been known as a bone-related hormone. Recently, however, it has been considered as an immune vitamin. Vitamin D3 deficiency influences the onset of a variety of diseases. Vitamin D3 regulates the production of proinflammatory cytokines such as tumour necrosis factor-α (TNF-α) through binding to vitamin D receptors (VDRs) in immune cells. Since blood levels of vitamin D3 (25-OH-D3) were low in coronavirus disease 2019 (COVID-19) patients, there has been growing interest in the importance of vitamin D3 to maintaining a healthy condition. On the other hand, phytochemicals are compounds derived from plants with over 7000 varieties and have various biological activities. They mainly have health-promoting effects and are classified as terpenoids, carotenoids, flavonoids, etc. Flavonoids are known as the anti-inflammatory compounds that control TNF-α production. Chronic inflammation is induced by the continuous production of TNF-α and is the fundamental cause of diseases like obesity, dyslipidaemia, diabetes, heart and brain diseases, autoimmune diseases, Alzheimer’s disease, and cancer. In addition, the ageing process is induced by chronic inflammation. This review explains the cooperative effects of vitamin D3 and phytochemicals in the suppression of inflammatory responses, how it balances the natural immune response, and its link to anti-ageing effects. In addition, vitamin D3 and phytochemicals synergistically contribute to anti-ageing by working with ageing-related genes. Furthermore, prevention of ageing processes induced by the chronic inflammation requires the maintenance of healthy gut microbiota, which is related to daily dietary habits. In this regard, supplementation of vitamin D3 and phytochemicals plays an important role. Recently, the association of the prevention of the non-disease condition called “ME-BYO” with the maintenance of a healthy condition has been an attractive regimen, and the anti-ageing effect discussed here is important for a healthy and long life.

## 1. Introduction

In nutrition science, carbohydrates, proteins, and fats are the three major nutrients. Vitamins are the fourth nutrient, minerals are the fifth, fibres are the sixth, and phytochemicals are the seventh nutrient. Vitamins other than vitamin D3 are not synthesised in the body; thus, they need to be consumed in foods. There are 13 types of vitamins in humans: the fat-soluble vitamins A, D, E, and K [1]; the water-soluble vitamins B 1–3, 5–6, 8–9, and 12; and vitamin C [2].

There are two types of vitamin D: plant-derived vitamin D2 (ergocalciferol) and animal-derived vitamin D3 (cholecalciferol). Vitamin D2 is mainly found in mushrooms, but its structure is slightly different from animal-derived vitamin D3. Because of the structural difference, many reports suggest that vitamin D3 is more effective in the human body because of its ability to bind with vitamin D binding proteins (VDBP) and the vitamin D receptor (VDR) [3]. Vitamin D3 is not only synthesised in skin cells but is also obtained from animal-derived food ingredients. Vitamin D3 is absorbed from the gastrointestinal tract when accompanied by oil, since it is fat-soluble, but the absorption ratio is reduced without oil [4,5].

Vitamin D3 is produced from a precursor of cholesterol, 7-dehydrocholesterol (provitamin D3), in skin cells after exposure to ultraviolet rays from sunlight. Vitamin D3 is known as a bone-related vitamin [6], enhancing calcium absorption from the intestine and increasing calcium levels in the blood. In the blood stream, vitamin D3 exists as 25-OH-D3 (calcidiol), and when the blood level of 25-OH-D3 drops, parathyroid hormone (PTH) is produced and calcium in the bones is released into the blood.

Recently, however, vitamin D3 has been considered as an immune vitamin that helps balance the immune response by controlling inflammation responses. It has been suggested that 25-OH-D3 deficiency in the blood is correlated with the severity of illness in respiratory infectious diseases such as influenza and coronavirus disease 2019 (COVID-19).

Vitamin D3 is fat-soluble and binds to proteins to circulate in the blood. Blood levels of vitamin D3 are measured using 25-OH-D3 levels. According to the international definition, a level of 30 ng/mL is considered sufficient, 29–20 ng/mL is insufficient, and 19 ng/mL or less is considered as a deficiency [7].

The half-life of 25-OH-D3 in circulation is only several weeks, and the active form is 1α,25-(OH)2-D3 (calcitriol), which is converted by the enzyme 25-hydroxyvitamin D 1-α-hydroxylase (cytochrome p450 27B1: CYP27B1) in the kidneys and immune cells [8]. CYP27B1 is tightly controlled by the function of fibroblast growth factor 23 (FGF23) and the Klotho gene (*KL*). The half-life of active vitamin D3 (1α,25-(OH)-2-D3) is only several hours, and it is converted to inactivated vitamin D3, 24(R),25-(OH)2-D3 (24,25-dihydroxycholecalciferol) by the cytochrome P450 family 24 subfamily A member 1 (CYP24A1) enzyme [9].

The term phytochemicals come from the Greek word for plant chemicals, and they are classified as terpenoids, carotenoids, flavonoids, sulphur-containing compounds, etc. Since they contain many phenolic hydroxyl groups, they are also called polyphenols and are attractive as beneficial nutrients for health [10]. β-glucan is also a type of phytochemical that works as a water-soluble dietary fibre and prebiotic (nutrient for gut microbiota).

Polyphenols used to be focused on due to their antioxidative effects, but recent research has been showing attractive anti-inflammatory properties. Phytochemicals that act as nutrients are found in a variety of vegetables and fruits, such as onions, citrus fruits, tea, soybeans, turmeric, cacao, and grapes, and over 7000 types are known.

In recent years, attention has been focused on the health-promoting effects of vitamin D3 and phytochemicals. This combined administration not only prevents “ME-BYO” (a non-disease condition), but also has anti-ageing effects due to the suppression of chronic inflammation via anti-inflammatory effects [11,12,13]. Until recently, even though the lifespan has increased, there were many people who could not enjoy healthy retirement since they continuously had to hospitalised. Recently, however, there has been increasing interest in healthy and long lives among people due to the prevention of lifestyle-related diseases caused by chronic diseases, even in the younger generation. This narrative review introduces the health-promoting effects, the prevention of ME-BYO, and the anti-ageing effects of vitamin D3 and phytochemicals derived mainly from vegetables and fruits, especially grapes.

## 2. Methods

This section briefly describes the method used to search for references in this review. This is a narrative review: information was collected using a PubMed search with complex keywords including “vitamin D3”, “phytochemicals”, “anti-ageing”, “gut microbiota”, “chronic inflammation”, and “immunity”. Furthermore, filtering functions were applied, such as searching only “review” articles and publication data limited within the last 5 years. The references cited were written in English and peer-reviewed papers. In addition to academic papers, this article also widely referred to press releases from universities and research institutes, as well as other scientific featured articles from newspapers and scientific information webpages and their reference papers. The current research referred to the latest references, except for references considered to be important, and over 40% of the references were limited to the last five years.

## 3. Vitamin D3

Vitamin D3 was named as the fourth vitamin by Elmer McCallum in 1922, and a Nobel prize laureate, Adolf Windaus, and others were associated with the early research. Vitamin D mainly consists of vitamin D2 (ergocalciferol) and vitamin D3 (cholecalciferol). Vitamin D3 is produced in skin cells exposed to ultraviolet (especially UVB) radiation. The vitamin D3 precursor provitamin D3 (7-dehydrocholesterol) is converted to previtamin D3 and then to vitamin D3. Vitamin D3 circulates in the blood stream after binding to vitamin D binding protein (VDBP) and is converted to 25-OH-D3 (calcidiol) by liver enzymes such as cytochrome P450 2R1 (CYP2R1). An enzyme, CYP27B1, in the kidneys and immune cells converts 25-OH-D3 to the active form, 1α,25-(OH)2-D3 (calcitriol), which binds to the vitamin D receptor (VDR) in the cytoplasm and translocates into the nucleus [14,15,16]. The complex of vitamin D-VDR stops the production of the cytokine, tumour necrosis factor-α (TNF-α), by binding to the promoter region of the TNF-α-producing gene (). In addition, this complex regulates the gene expression of calcium transporter proteins [17]. Most cells express VDRs, including cells in the brain, heart, skin, reproductive organs, prostate, and breasts, and white blood cells such as monocytes, activated T cells, and B cells [18]. The activated form of vitamin D3 (1α,25-(OH)2-D3) is inactivated to 24(R),25-(OH)2-D3 (24,25-dihydroxycholecalciferol) within a few hours by the enzyme CYP24A1. Figure 1 shows the pathway of vitamin D3 synthesis, from provitamin D3 to the activated form (1α,25-(OH)2-D3), and inhibition of TNF-α production.

The amount of vitamin D3 is shown as the blood concentration of 25-OH-D3 (1 IU = 25 ng (40 IU = 1 μg); 1 mol = 0.4 ng/mL). According to the international standards, a blood concentration of 30 ng/mL or higher is considered sufficient, 20–29 ng/mL is insufficient, and 19 ng/mL or less is considered deficient [7].

The blood concentration of 25-OH-D3 varies depending on the latitude of a place and the season. This is because the intensity and time of exposure to UV light differ and are low in winter when the altitude of the sun is low. Since vitamin D3 is produced by skin cells, ethnic differences exist in its production, as the pigment melanin in the skin blocks UV light [19]. People with light skin can produce enough vitamin D3 after exposure to sunlight; however, if the skin is shielded from UV light or there is an accumulation of melanin in the skin, they cannot produce sufficient levels of vitamin D3.

Rickets was a major public health concern in the United States during the first half of the 20th century, but has nearly been eradicated with the introduction of vitamin D-fortified milk, paediatric supplementation of vitamin D3, and the increased intake of animal proteins. In the 21st century, however, rickets is found only in low-income countries in Africa, Asia, and the Middle East or regions with genetic predispositions to vitamin D3 deficiency [20,21].

In the UK, people at risk of vitamin D3 deficiency are recommended to take a daily supplementation of vitamin D3, and 10 μg (400 IU)/day is the recommended dose from autumn to the winter months [22]. In the U.S., the 2016 recommended dietary allowance (RDA) was around 20 μg (800 IU)/day, and similar standards have been set in Canada, Australia/New Zealand, and the EU. According to the European Food Safety Authority (EFSA), the permissible upper limit for adults is 100 μg (4000 IU)/day, which is the same as the limit in the U.S. [23,24,25].

Normally, people absorb 60–80% of ingested calcium from the diet in the intestine, but people in vitamin D3 deficiency absorb only 15% of normal levels of calcium [26,27]. When the blood 25-OH-D3 concentration falls lower than 19 ng/mL, the risks of fracture and frailty increase. Osteoporosis is an adult disease induced by long-term vitamin D3 deficiency that results in the risk of spinal curvature and fractures [28]. Vitamin D3 also mitigates the risk of bone density loss induced by cadmium poisoning [29].

### 3.1. Vitamin D3 and Infectious Diseases

In the past, sunbathing was used as a treatment for tuberculosis based on experience. Later, it was confirmed that vitamin D3 enhances the function of macrophages in the lungs and increases the production of the antimicrobial peptide cathelicidin (LL-37). In addition, vitamin D3 increases the production of another antimicrobial peptide, β-defensin, and helps improve the intestinal environment [30].

After infections by pathogens, macrophages produce proinflammatory cytokine TNF-α and induce inflammatory responses. In this context, vitamin D3 stops the production of TNF-α and suppresses its responses to avoid inducing excess or chronic inflammation [31].

The onset of seasonal flu is associated with serum 25-OH-D3 levels. Influenza commonly manifests in winter and coincides with the period when vitamin D3 synthesis due to UV irradiation is low. In addition, the onset of COVID-19 is associated with low blood 25-OH-D3 levels.

Ilie et al. indicated that COVID-19 deaths were higher in Spain, Italy, Switzerland, Belgium, and the United Kingdom, where the average serum 25-OH-D3 levels were deficient (>20 ng/mL) [32]. The data from Johns Hopkins University showing the numbers of infected people (with % mortality), as of March 10, 2023, are as follows: ① United States—103,802,702 (1.08%), ② India—44,690,738 (1.19%), ③ France—38,618,509 (0.42%), ④ Germany—38,249,060 (0.44%), ⑤ Japan—37,320,438 (0.20%), ⑧ Italy—25,603,510 (0.74%), ⑨ UK—24,425,309 (0.90%), and ⑫ Spain—13,770,429 (0.87%). The blood 25-OH-D3 levels were: ③ France—24.0 (ng/mL), ④ Germany—20.0, ⑧ Italy—20.0, ⑨ UK—19.0, and ⑫ Spain—17.0.

According to a study conducted by Miyamoto et al. (2023), the average serum 25-OH-D3 level in Japanese people was 15.5 ng/mL in a total of 5518 Japanese people (3400 men and 2118 women). Most of them were deficient (>19 ng/mL), and people with enough serum 25-OH-D3 (<30 ng/mL) represented only 2% of the population sample. The number of infected people was the fifth highest, as shown above, but the mortality ratio was lower than in other countries. This result is probably due to the differences in the medical environments between Europe, America, and Japan. Treating COVID-19 with 25-OH-D3 is being considered, and it might alleviate the severe symptoms of COVID-19. Some clinical research shows that vitamin D3 is more effective than vitamin D2 [33].

The concomitant intake of excess amounts of vitamin D3 and calcium induces hypercalcaemia, leading to liver dysfunction and kidney damage [34,35]. As vitamin D3 is a bone-related vitamin, vitamin D3 and calcium tend to be co-administered at the same time. Therefore, with increased amounts of vitamin D3, the incorporation of calcium increases, leading to hypercalcaemia. However, as vitamin D3 is considered an immune vitamin, daily intake of only vitamin D3 to maintain blood levels > 30 ng/mL is highly recommended, rather than concomitant intake with calcium. Research shows that only a few countries have average blood levels of 25-OH-D3 over 30 ng/mL.

There is a possibility that a natural immune response eliminates the virus in people with asymptomatic infections or mild symptoms of COVID-19. Vitamin D3 is known to affect both mucosal and natural immunity [36,37], and it is expected that vitamin D3 work as a preventive measure against viral infections. It is expected that maintaining > 30 ng/mL of serum 25-OH-D3 levels prevents the onset of severe COVID-19.

Continuous intake of 50 times the recommended amount of vitamin D3 (1250 μg (50,000 IU)/day) induces vitamin D toxicity within several months, and blood levels of activated vitamin D3 (1α,25-(OH)2-D3) reach 150 ng/mL in some cases. The abnormally high levels of vitamin D3 recover after the excessive intake is stopped.

### 3.2. Vitamin D3 and Natural Immunity

Fibroblast growth factor (FGF23) is an important factor for regulating the metabolism of phosphorus and vitamin D3, and excess FGF23 induces hypophosphataemia and low serum 25-OH-D3. Conversely, a lack of FGF23 induces hyperphosphataemia and high serum 25-OH-D3. Mice with a Klotho gene mutation show diverse phenotypes like human ageing from a young age and have shorter lives [38]. In addition to the Klotho gene mutation, mice with FGF23 deficiency have shorter lifespans, high serum 25-OH-D3, hyperphosphataemia, and hypercalcaemia. A vitamin D3-restricted diet improves these conditions. The FGF23/Klotho system works as a negative feedback mechanism for vitamin D3. The CYP27B1 enzyme, which adds a hydroxyl group to the 1α position of 25-OH-D3, is the rate-limiting enzyme, and the activated 1α,25-(OH)-D3 binds to the VDR together with the retinoid X receptor (RXR). This complex is involved in the regulation of at least 1000 genes in the human genome [39].

For the natural immune response, *CYP27B1* gene expression in mice is induced by stimulation from lipopolysaccharide (LPS) and interferon-γ (IFN-γ). Modlin et al. indicated that the stimulation of human macrophages with toll-like receptor 2 (TLR2) ligands induced the expression of *VDR* and *CYP27B1* genes, increased the production of active vitamin D3 (1α,25-(OH)2-D3), and decreased the production of the proinflammatory cytokines TNF-α and interleukin-6 (IL-6) [40]. This phenomenon suggests that high serum 25-OH-D3 levels stimulate cells in the innate immune system and that activated vitamin D3 (1α,25-(OH)2-D3) suppresses the inflammation. Neutrophils express the same VDR mRNA as macrophages, and gene expression of *CD14* and cathelicidin antimicrobial peptide (CAMP)/LL-37 (*CAMP*) is observed in the presence of activated vitamin D3 (1α,25-(OH)2-D3) [41]. This suggests the involvement of vitamin D3 in neutrophil activation during bacterial infections. Activation of the vitamin D3 signalling pathway occurs in natural killer (NK) cells [42].

Furthermore, CAMP/LL-37 (*CAMP*) and β-defensin 2 (*DEFB4A*) genes are located adjacent to vitamin D response element (VDRE).

During viral infections, type-1 interferon (IFN), IFN-α, and IFN-β exert antiviral activity during viral infections, and CAMP/LL-37 exerts antiviral activity as well [43]. Vitamin D3 acts protectively during infections with several viral strains, including hepatitis viruses, the human immunodeficiency virus, and respiratory pathogenic viruses.

### 3.3. Vitamin D3 and Chronic Inflammation

In addition to bone-related diseases induced by vitamin D3 deficiency, vitamin D3 has been shown to work as an immune vitamin [44,45,46,47,48,49]. In obesity, free fatty acids released from adipocytes bind to toll-like receptor 4 (TLR4), promoting proinflammatory cytokine TNF-α, and induce chronic inflammation. A series of signal transduction pathways, including mitogen-activated protein kinase kinase kinase 3 (MAP3Ks), which are involved in cell proliferation and death, are located downstream of TLR4 [50]. Inflammation in blood vessels induces arteriosclerosis, and inflammation in the liver initiates non-alcoholic fatty liver disease (NAFLD). VDRs are distributed in the cells in the small intestine, and α-defensin produced by Paneth cells promotes natural immunity. Therefore, vitamin D3 deficiency reduces α-defensin production and impairs the natural immune function of the small intestine. α-Defensin deficiency is also associated with depression, and stress reduces the functionality of tight junctions and induces bacterial translocation in the body. A decrease in tight junctions between intestinal epithelial cells, called leaky gut syndrome, induces an influx of endotoxins and bacteria in the blood stream and induces chronic inflammation. In addition, chronic inflammation caused by vitamin D3 deficiency induces metabolic syndrome and insulin resistance. NAFLD is more likely to have its onset when serum 25-OH-D3 levels are low. It is reported that a VDR gene single-nucleotide polymorphism (SNP) is also associated with liver fibrosis [51].

In the ageing process, the number of senescent cells increases, and aged macrophages release senescence-associated secretory phenotype (SASP) factors. Chronic inflammation induced by SASP factors induces ageing-associated diseases such as cardiovascular disease, arteriosclerosis, cancer, diabetes, chronic kidney disease, non-alcoholic steatohepatitis (NASH), autoimmune diseases, and neurodegenerative diseases. Vitamin D3 deficiency accelerates chronic inflammation induced by SASP factors via cellular senescence. After menopause, osteoclast activity controlled by the female hormone oestrogen is activated, leading to osteoporosis, a condition of decreasing bone density [52].

Vitamin D3 deficiency reduces the absorption of calcium from the digestive tract, and the intake of vitamin D3 and vitamin K is important for this pathway [53]. In addition, a soybean isoflavone genistein is converted to equol by the gut microbiota, and it works as a phytoestrogen. In Europe and America, only one third of people have equol-converting gut microbiota; hence, equol is not produced, even after the consumption of genistein. The consumption of fermented foods is associated with the number of equol-producing gut microbiota [54].

### 3.4. Vitamin D3, Gut Microbiota, and Gut Environment

The gut environment is affected by what we eat. Characteristic gut microbiomes are formed by the regional and ethnic differences. In addition, there are differences between the genders because of the difference of sex hormones. Furthermore, these differences are caused by the intake of fermented foods, and gut microbiota are different between Japanese and Western people as well.

After mice were fed a high-fat diet, *Bacteroidetes* decreased dramatically and *Firmicutes* and *Verrucomicrobium* increased in the high-fat-diet group compared to the normal-diet group [55]. However, the intake of a high-fat diet in humans indicated the complete differences between mice and humans. In European and American experiments comparing diabetic and healthy subjects, *Roseburia intestinalis* and *Faecalibacterium prausnitzii* in the phylum *Firmicutes* were decreased in diabetic subjects [56], and *Bacteroidetes* and *Proteobacteria* were increased in normal subjects [57]. In addition, in a comparison of healthy and type-2 diabetic Japanese subjects, different results were observed, with a decrease in *Bacteroidetes* and increases in *Firmicutes* and *Actinobacteria* in diabetic subjects [58]. Therefore, it is important to recognise the differences in gut microbiota depending on the different ethnicities and dietary habits.

Many reports have supported the relationship between vitamin D3 and gut microbiota. Vitamin D3 intervention research has shown increases in health-promoting *Ruminococcaceae*, *Akkermansia*, *Faecalibacterium*, and *Coprococcus* bacteria and a decrease in *Firmicutes* [59,60].

Not only can vitamin D3 enhance calcium absorption, VDRs are also expressed in the cells in the digestive tract and maintain the immune system in the gut. Vitamin D3 deficiency is correlated with the onset of inflammatory bowel disease (IBD), as a decrease in *Faecalibacterium prausnitzii* was observed in patients with IBD [61].

Changes in gut microbiota composition across generations have been observed. Odamaki et al. indicated that the characteristic gut microbiota of centenarians in Japan’s Amami Islands had increased *Bifidobacterium*, *Akkermansia*, and *Metathnobrevibacter* [62]. Johansen et al. indicated that the gut microbiota in some centenarians can decompose sulphurs and help them to stay healthy [63]. In addition, there is a difference in the gut microbiota even between adults and elderly people [64].

The bile acids cholic acid (CA) and chenodeoxycholic acid (CDCA) are produced from cholesterol in the liver. These primary bile acids form conjugates and are secreted into the duodenum. Bile acids secreted into the digestive tract modify lipids into micelles and make them susceptible to decomposition by lipase. Most bile acids are circulated to the liver by transporters (enterohepatic circulation); however, some of them circulate throughout the body and exert physical activities in organs and tissues. Bile acids have three different effects on gut microbiota: (1) bacterial sterilisation and bacteriostatic effects, (2) germination, and (3) suppression of pathogenic gene expression. In addition, bile acids work as ligands for the nuclear receptor farnesoid X receptor (FXR) and the cell membrane receptor Takeda G protein-coupled receptor 5 (TGR5). Both receptors are expressed in the liver, digestive tract, pancreas, white and brown adipocytes, mononuclear phagocytic dendritic cells, Kupffer cells, and macrophages and influence the physiological functions of the host. Sato et al. indicated characteristic bile acids in centenarians—iso-lithocholic acid (LCA), 3-oxo-LCA, and iso-allo-LCA—which are produced by specific strains of gut microbiota in healthy centenarians [65]. Figure 2 summarises the cellular distribution of bile acid receptors FXR and TGR5 when expressed in the body.

Bile acids bind to TGR5 in the digestive tract and induce intestinal peristalsis. Peristalsis decreases with ageing, which tends to become a cause of constipation. The wax component, called bloom, on the surfaces of grapes and other plants is the TGR5 agonist triterpene oleanolic acid. Recently, searches for the TGR5 agonist have been conducted across the world. Fermented grape food from Koshu (K-FGF) contains oleanolic acid and tends to relieve constipation after its consumption [66,67,68].

## 4. Phytochemicals

As we have reported previously [66,67,68,69,70], terpenoids, carotenoids, flavonoids, and even β-glucans are included in phytochemicals, a variety of plant-derived chemicals. Recently, research into chemicals derived from plants has also been conducted to develop safe alternatives to petroleum-derived industrial chemicals that might have adverse effects on the environment and its residents [71]. Here, this review describes the phytochemicals found in vegetables and fruits that have health-promoting effects. Polyphenols in grapes are one of the most researched phytochemicals. The attention to grape phytochemicals started with the red wine polyphenol resveratrol, which gained the attention due to the “French Paradox”—the mortality rate from cardiovascular disease in French individuals is one-third that of Americans. In addition, the Mediterranean diet, which is considered as healthy around the world, contains many phytochemicals [72].

Flavonoid polyphenols are listed: quercetin (onion) [66], catechins (green tea and wine) [73], theaflavin (black tea) [74], anthocyanin [75], hesperidin (citrus) [76], isoflavone (soybeans) [77], sesamin (sesame) [78], etc. Non-flavonoid polyphenols include curcumin; chlorogenic acid (coffee); caffeic acid; ferulic acid (rice bran) [79]; allicin (garlic, leek, and green onion) [80]; and carotenoids such as α-carotene, β-carotene, and β-cryptoxanthin [81]. The intake of foods with antioxidant properties is effective for the removal of reactive oxide in the blood. Furthermore, research on chemicals with anti-inflammatory properties for the prevention of diseases associated with chronic inflammation is increasing. Grapes contain a variety of phytochemicals, including anthocyanins, quercetin, catechins, caffeic acid, and carotene [67].

### 4.1. Terpenoids

Terpenoids are chemicals with a triterpene skeleton structure, including oleanolic acid, ursolic acid, and saponin (sugar attached). Oleanolic acid is a component of the white powder called bloom on the surfaces of grapes, which works as an agonist of the bile acid receptor TGR5 on the cell surface [82,83]. Since oleanolic acid also works as an agonist of nuclear receptor FXR, it activates genes and transcriptional networks associated with the metabolism of sugar and lipids, energy consumption, and inflammation [84]. Furthermore, oleanolic acid plays an important role in vitamin D3 metabolism. Oleanolic acid enhances CYP27B1 for the generation of the active form of vitamin D3 (1α,25-(OH)2-D3) and CYP24A1 for the decomposition of the 1α,25-(OH)2-D3. Therefore, oleanolic acid has an important role in bone metabolism. Ursolic acid, another triterpene chemical, is also effective in skeletal muscle health, along with vitamin D3 [85].

### 4.2. Carotenoids

Well-known carotenoids include β-carotene, astaxanthin, β-cryptoxanthin, etc. The main carotenoids contained in grapes are β-carotene and lutein. β-Carotene is a vitamin A precursor mainly obtained from food intake; hence, grape phytochemicals are also helpful for the maintenance of vision [66,86,87].

### 4.3. Flavonoids

The flavonoids contained in grapes include catechins, flavan-3-ols, flavon-3-ols, etc. The most abundant procyanidins in grapes are oligomeric procyanidins, the complexes of (epi)catechins. Furthermore, quercetin, a flavone-3-ol, is the second most abundant phytochemical in grapes after catechins, and it also exists in conjugates like quercetin–glycoside and quercetin–glucuronide [88,89,90]. Table 1 summarises the variety and effects of polyphenol flavonoids shown in this review.

### 4.4. Attempts to Improve the Bioavailability and Activation of Phytochemicals in the Body

Low bioavailability of the phytochemicals is well known problem. Generally, phytochemicals have low solubility in the water, making it difficult for them to reach their target organs and cells. Therefore, the application of nanotechnology has been an attractive choice from the perspective of a drug delivery system (DDS) for phytochemicals, as they reach their targets, especially in cancer chemotherapy. Efforts are being made to deliver phytochemicals into specific organs or cells by creating capsules using nanotechnology. This attempt is being conducted with high expectancy [93,94].

Another aspect is the chemical modification of phytochemicals. Compounds like hesperidin and quercetin are insoluble in the water by themselves; however, after the glycosylation, they become water-soluble and reach the intestines [95]. Thus, they are broken down by enzymes from gut microbiota to hesperitin and quercetin monomer and can pass through the intestinal cell wall. After absorption from the intestines, quercetin conjugates with glucuronide with the effect of the enzyme and is absorbed into the bloodstream [96]. In atherosclerosis, quercetin–glucuronide is taken up by the foam cells in blood vessels and becomes active after the removal of glucuronide by the enzyme secreted in form cells. Activated quercetin manifests the preventative effects and suppresses the formation of thrombosis [97].

### 4.5. Consideration of the Affinity of Phytochemicals to the VDR

Other researchers have suggested the direct binding ability of phytochemicals to the VDR. For instance, curcumin, a phytochemical contained in turmeric, binds to the VDR as a ligand and exhibits physiological roles [98,99,100]. Moreover, the combination of vitamin D3 and curcumin has been reported to prevent ageing of the brain and maintain healthy conditions [101]. According to the latest research, the flavonoid quercetin also interacts with VDR and prevents breast cancer and fibrosis, participating in anti-ageing [91].

## 5. Synergy between Vitamin D3 and Phytochemicals

The ageing process is accelerated by the accumulation of chronic inflammation. Various diseases initiated by chronic inflammation promote continuous TNF-α production. For this reason, during a COVID-19 infection, TNF-α production increases explosively and induces a cytokine storm. On the other hand, regular intake of phytochemicals and maintenance of high blood 25-OH-D3 levels upregulate natural immune responses and induce latent infections or mild conditions, even when infected.

When receptors of macrophages are stimulated by ligands, the receptors aggregate and form a raft on the cell membrane. Then, signals are transmitted into the cell and activate a transcription factor, nuclear factor-κ B (NF-κB) [102,103]. Phytochemicals, especially flavonoids, exhibit anti-inflammatory effects through the downregulation of signal transduction and stop the activation of NF-κB. For example, the flavonoid quercetin inhibits the signal transduction by inhibiting raft formation. Figure 3 shows the inhibitory mechanism of TNF-α gene (*TNF*) expression via raft formation by flavonoids and LL-37. Flavonoids and LL-37 also stop the production of proinflammatory cytokines IL-1β and IL-6. In addition, activated vitamin D3 (1α,25-(OH)2-D3) binds to the VDR and stops the expression of the *TNF* gene by binding to the promoter region, as shown in Figure 1. Thus, the interaction between the vitamin D3 and phytochemicals inhibits the production of TNF-α and stops the progression of ageing via the prevention of chronic inflammation.

### 5.1. Vitamin D3 and Phytochemicals in Bone Metabolism

Oleanolic acid, a triterpene, is an important phytochemical that enhances healthy bone metabolism by increasing the production of the enzyme CYP27B1, which converts 25-OH-D3 to an active form of vitamin D3 (1α25-(OH)2-D3), and suppressing the expression of the enzyme CYP24A1, which inactivates the 1α25-(OH)2-D3. Oleanolic acid works as an agonist of the bile acid receptor TGR5 in the intestine, increasing intestinal peristalsis. TGR5-expressing intestinal L cells and pancreatic α cells increase the expression of glucagon-like peptide-1 (GLP-1), and pancreatic β cells enhance insulin secretion, preventing the onset of diabetes [104]. TGR5 in white and brown adipocytes increases energy consumption and prevents the manifestation of obesity. In addition, oleanolic acid suppresses inflammation by suppressing the production of interleukin-12 (IL-12) and TNF-α in dendritic cells [105]; cytokine production in Kupffer cells; and production of proinflammatory cytokines TNF-α, interleukin-1β (IL-1β), and IL-6 in macrophages [106].

During menopause, the decrease in the female hormone oestrogen increases susceptibility to osteoporosis. The effects of phytoestrogen equol and the osteoporosis preventive effect of β-cryptoxanthin have already been mentioned above [68]. Research into the effects of the combined administration of vitamin D3 and tea catechins has been conducted for the inhibition of bone metabolism by osteoclasts [107]. *Citrus unshiu* mandarin oranges contain large amounts of the carotenoid β-cryptoxanthin, and research by the National Agriculture and Food Research Organisation (NARO) of Japan has analysed their ability to prevent the onset of osteoporosis [108].

### 5.2. Vitamin D3 and Phytochemicals in the Prevention of Ageing

Mitochondria are cellular organelles responsible for producing ATP, a source of energy. With the increase in senescent cells due to ageing or poor living conditions, the number of mitochondria decreases, and the release of reactive oxide species from aged mitochondria increases [109,110]. Cells are rejuvenated by the removal of aged mitochondria and organelles through autophagy when they are no longer needed.

Cellular metabolism of the energy is controlled by the body clock, but the ageing process deteriorates this clock [111,112]. A low-calorie diet restores body clock malfunctions due to the ageing [113]. Calorie restriction activates the sirtuin genes, the longevity related genes rejuvenating cells in the body. The sirtuin genes vary from *SIRT1* to *SIRT7* [114,115]. In Japan, it is encouraged for the people to limit their dietary intake to 70–80% of a full stomach, which is approximately a 30% calorie restriction. The regeneration of mitochondria requires the activation of a protein termed peroxisome proliferator-activated receptor γ coactivator 1-α (PGC1α). This process is controlled by AMP-activated protein kinase (AMPK), an enzyme that does not work when the stomach is full [116]. The activation of sirtuin genes plays significant roles in several diseases, including the inhibition of neurodegenerative diseases and dementia, the protection of the myocardium, and the improvement of liver metabolism. Furthermore, sirtuin genes prevent the body from accumulating fat by promoting insulin secretion, thus improving skeletal muscle metabolism. Hence, this activation suppresses dementia, age-related hearing loss, fatty liver, cancer, and cardiovascular diseases [117,118]. The anti-ageing mechanism of the sirtuin genes prompts ageing cells to repair DNA, restoring the activity of cells and rejuvenating cells throughout the body [119]. This process leads to improvements in skin blemishes, wrinkles, skin barrier function, ceramide synthesis, and muscle and liver function [120].

The senescence of cells is caused by epigenetic changes induced by oxidative stress, such as X-rays, ultraviolet rays, and certain drugs, which have adverse effects on the chromosomes. In normal cells, the number of cell divisions is determined by telomerase, and cells undergo cell death. However, in senescent cells, they release SASP factors and induce inflammation instead of undergoing cell death [121]. Since chronic inflammation leads to fibrosis in several organs, such as the lungs and liver, the removal of senescent cells is important in order to delay fibrosis [122]. The grape polyphenol resveratrol first gained attention for the activation of sirtuin genes; however, nicotinamide mononucleotide (NMN) has recently been recognised as an effective substance [123,124]. In addition, oligomeric procyanidins found in grapes also contribute to the activation of sirtuin genes [125].

As mentioned above, the Klotho gene is very critical, and its deficiency leads to premature death. In addition, along with the *FGF23* gene, this gene is necessary for the expression of the CYP27B1 enzyme, which converts 25-OH-D to the active form of vitamin D3, 1α,25-(OH)2-D3 [38]. The expression of the Klotho gene decreases with age, and research on mice overexpressing the Klotho gene shows the extension of their lifespans. Hence, the importance of the Klotho gene has been recognised. Furthermore, it is of particular interest that vitamin D3 metabolism is associated with the onset of a variety of diseases. Therefore, vitamin D3 and phytochemicals have a synergistic effect on anti-ageing by activating ageing-suppressing genes.

### 5.3. Vitamin D3 and Phytochemicals in the Suppression of Chronic Inflammation

Chronic inflammation causes obesity, hypertension, diabetes, and dyslipidaemia. Furthermore, arteriosclerosis induces cardiovascular diseases such as myocardial infarction and stroke, autoimmune disease, Alzheimer’s disease, and cancer. Non-alcoholic fatty liver disease (NFALD) and lung disorders like interstitial pneumonia and chronic obstructive pulmonary disease (COPD) are also examples of diseases induced by chronic inflammation [67]. Advanced glycation end-products (AGEs), resulting from the Maillard reaction between sugars and proteins, are typical ageing-related substances that accelerate the ageing process. The Maillard reaction, a chemical glycation process between the sugar and amino groups of proteins, produces glycated proteins instead of glycosylation of the proteins [126]. Flavonoids such as quercetin, luteolin, and rutin suppress the production of AGEs, and AGEs stimulate macrophages to produce the proinflammatory cytokine TNF-α. Flavonoids suppress the TNF-α production induced by AGEs and, therefore, suppress ageing [127]. AGEs induce inflammation in the brain, and the suppression of glucose uptake by the inflammation induces both cell death and diseases like Alzheimer’s disease [128]. The chronic inflammation induced by AGEs causes the glycation of bone collagen, making bones easier to fracture, and increases skin collagen stiffness. The intake of vitamin D3 and flavonoids improves quality of life (QOL), suppresses ageing, and is associated with a long and healthy life [129].

Periodontal disease is not only a risk factor for several diseases like diabetes, cardiovascular disease (atherosclerosis), aspiration pneumonia, premature birth and low birth weight, bacterial heart disease, sepsis, glomerulonephritis, and arthritis, but is also associated with Alzheimer’s disease [130]. In a cohort study, Alzheimer’s disease patients were divided into two groups based on the presence or absence of periodontal disease. After six months, a significant decrease in cognitive function was observed in the patients with periodontal disease. Elevated systemic inflammation levels are associated with the production of inflammatory mediators in the meninges, activating microglia in the brain parenchyma. LPS from periodontal bacteria increased the accumulation of amyloid in the brain [131]. This suggests the possibility that periodontal disease bacteria or pathogenic factors infiltrate into the brain and aggravate inflammation or that systemic inflammation caused by periodontal disease exacerbates brain inflammation. The management of diseases induced by chronic inflammation is crucial for slowing down the ageing process in individuals. In addition, flavonoids help with managing septic shock, which is induced by infectious diseases [92,132].

In the Mediterranean diet, which is known as a healthy diet worldwide, red meat is only consumed about once a week; however, recently, with the influence from the United States, the consumption of meat was recommended in Japan. The consumption of a high-fat diet has a negative effect on the gut microbiota and induces visceral obesity and NAFLD. Palmitic acid, a saturated fatty acid found in pork and beef, induces TNF-α production through TLR4, contributing to the development of diabetes and atherosclerosis. Periodontal disease, which affects 80% of people over the age of 30, is attributed to TNF-α production induced by periodontal bacteria. Respiratory diseases, osteoporosis, heart disease, and diabetes are also associated with periodontal diseases [133]. After menopause, the reduction in oestrogen leads to increases in the proliferation of osteoclasts and bone resorption activity, resulting in the onset of osteoporosis. It is well known that vitamin D3 participates in the prevention of osteoporosis [134,135]. The upregulation of TNF-α production by accumulation of AGEs is also suppressed by vitamin D3 and phytochemicals. Therefore, this interaction is effective in anti-ageing through the suppression of chronic inflammation. Figure 4 summarises the anti-ageing mechanisms involving vitamin D3 and phytochemicals.

### 5.4. Foods with Anti-Ageing Effects

The Mediterranean diet is an example of a representative dietary intervention for anti-ageing. Although it is called the Mediterranean diet, exercise is included as a fundamental activity [136]. The diet consists of low-glycaemic-index (GI) whole grains (bread and pasta), vegetables, fruits, olive oil, fish, and wine, with reduced intake of red meat. As a result, the incidences of dyslipidaemia, diabetes, coronary artery disease, and hypertension, as well as the risk of Alzheimer’s disease, decrease [137,138,139]. Vegetables and fruits also contain large amounts of phytochemicals [140,141]. Extra-virgin olive oil, rich in the ω-9 polyunsaturated fatty acid oleic acid, especially lowers low-density lipoprotein (LDL) cholesterol and triglyceride (TG), contributing to a reduction in cardiovascular disease [142].

The Japanese diet is being recognised for its association with longevity. The diet that leads to a healthy and long life is the diet which was consumed around 1975 in Japan [143]. In the generation of those aged 40 and below, there is a tendency to skip breakfast and favour an American-styled diet. For this reason, recent research has shown that the condition of the gut microbiota of this generation is less favourable than those of groups of people eating a Japanese diet or a Westernised but healthy diet. Calorie restriction by eating until 80% full activates the sirtuin gene, leading to a healthy and long life. The consumption of fish, seafood, and phytochemicals rather than red meat is linked to good health. The water-soluble dietary fibres found in glutinous barley and Jerusalem artichokes serve as nutrients for butyric-acid-producing bacteria in the gut and help prevent constipation, which increases with age. Oleanolic acid, a triterpene, is an agonist of the bile acid receptor TGR5, and it helps with intestinal peristalsis. It is also known to enhance glucan synthase-like1 (GSL1)-mediated effects [144]. Oleanolic acid is a substance found in vegetables and fruits, especially in grape skin.

Since folic acid (vitamin B9) has been shown to help prevent dementia [145,146], experiments with the administration of vitamin D3 and folic acid in mild Alzheimer’s disease were conducted. The consumption of foods rich in vitamin B12 and folic acid suppressed hyperhomocysteinaemia and prevented arteriosclerosis and dementia. Hyperhomocysteinaemia is a risk factor for Alzheimer’s disease, and according to the research, it has been shown that Japanese people are often genetically predisposed to folic acid deficiency.

As this review has shown above, the supplementation of vitamin D3 and phytochemicals from vegetables and fruits is an effective strategy for anti-ageing. In addition, the ω-3 fatty acids eicosapentaenoic acid (EPA) and docosahexaenoic acid (DHA) have important roles in the suppression of chronic inflammation. DHA is a major component of the fatty acids that make up the phospholipids in the brain. DHA softens the membranes of nerve cells, increases cellular processes, and promotes axonal growth, thereby maintaining the health of the brain and nervous system [147]. Regions consuming more seafood have higher concentrations of DHA in breast milk and lower prevalence of postpartum depression [148].

Recently, finding “ME-BYO”, a non-disease condition, and treating it before the manifestation of diseases to achieve healthy and long life has been common in Japan, which is a typical stressful society. This term originated from the ancient Chinese medical textbook “*Huangdi Neijing*” about 2000 years ago. The continuous implementation of ME-BYO measurement from a young age makes it possible to extend the healthier life expectancy and reduce medical expenses [149,150,151]. To avoid being ME-BYO, sleeping well, performing moderate exercise, and eating a healthy diet are important. Furthermore, the prevention of ME-BYO is not only good for healthy and long life, but also good for anti-ageing.

## 6. Conclusions

In conclusion, the intake of vitamin D3 and phytochemicals found in vegetables and fruits is beneficial for the prevention of both ageing and “ME-BYO” (a non-disease condition). In the ageing process, there is concern about the decline in the muscles supporting the body’s skeletal structure, leading to mental and social declines. To achieve a long and healthy life, it is important to maintain one’s basal metabolism through exercise and diet and to prevent mental illnesses that lead to a loss of social skills. This is effective to prevent ME-BYO as well. Vitamin D3 deficiency causes a variety of diseases, and the intake of phytochemicals is associated with the upregulation of natural immunity and reductions in the risks of various diseases. The simultaneous intake of vitamin D3 and grape phytochemicals is recommended because this enhances their anti-ageing effects and prevents the ME-BYO condition, leading to a long and healthy life. Recently, attention paid to healthy longevity has been increasing to an unprecedented level. Hence, it is expected that research into the prevention of lifestyle-related diseases and ageing-related diseases caused by chronic inflammation through the intake of vitamin D3 and phytochemicals will continue increasing in the future.

## Figures and Tables

**Figure 1 ijms-25-02125-f001:**
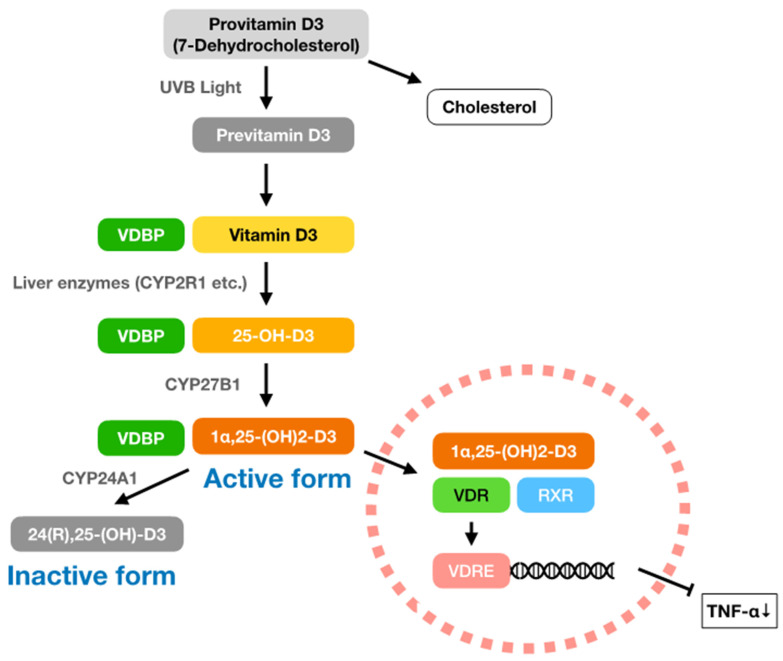
Suppressive pathway of TNF-α production by vitamin D3. Within the vitamin D, animal-derived vitamin D3 (cholecalciferol) is made from provitamin D3 (7-dehydrocholesterol), which is also a precursor of cholesterol. Provitamin D3 is converted to previtamin D3 in the skin cells by UVB light, and previtamin D3 becomes vitamin D3. After being conjugated to vitamin D binding protein (VDBP), vitamin D3 circulates into systemic blood stream and is then converted into 25-OH-D3 (calcidiol) by the liver enzymes cytochrome P450 2R1 (CYP2R1), etc. Blood levels of vitamin D3 are evaluated according to 25-OH-D3 concentration and >30 ng/mL is considered as a sufficient amount. An enzyme 25-hydroxyvitamin D 1-α-hydroxylase (cytochrome p450 27B1: CYP27B1) exists in the kidneys and immune cells, which convert 25-OH-D3 to 1α,25-(OH)2-D3 (calcitriol). This activated form of vitamin D3 (1α,25-(OH)2-D3) binds to retinoid X receptor (RXR) and forms a complex with vitamin D receptor (VDR). Finally, vitamin D3-VDR complex translocates into the nucleus and binds to vitamin D response element (VDRE), including the promoter region of TNF-α gene (*TNF*), and stops the production of TNF-α. Thus, total amount of TNF-α in the body decrease. Active form of vitamin D3 (1α,25-(OH)-D3) is converted to inactive form, 24(R),25-(OH)-D3 (24,25-dihydroxycholecalciferol), by an enzyme cytochrome P450 family 24 subfamily A member 1 (CYP24A1).

**Figure 2 ijms-25-02125-f002:**
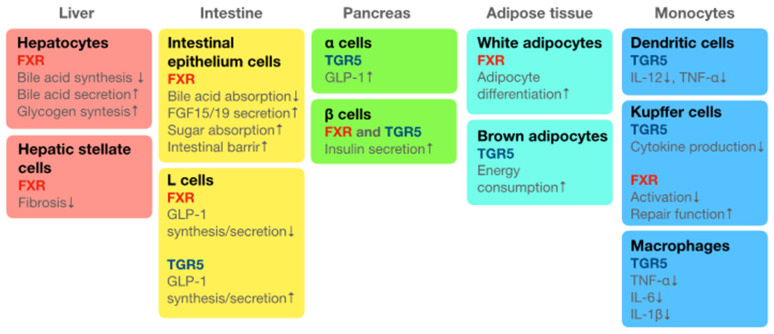
Summary of the distribution of bile acid receptors TGR5 and FXR when expressed in the body. Oleanolic acid is the agonist of bile acid. Takeda G protein-coupled receptor 5 (TGR5) and farnesoid X receptor (FXR) are the receptors of bile acid systemically distributed in the body: TGR5 is distributed on the cell surface and FXR is distributed in the nucleus. Current review summarises the distribution of TGR5 and FXR in each organ or tissue at the cellular level. This figure also summarises the physiological functions of bile acids and their agonist, oleanolic acid, through binding to TGR5 or FXR. Down arrows (↓) indicate decrease/downregulation and up arrows (↑) indicate increase/upregulation.

**Figure 3 ijms-25-02125-f003:**
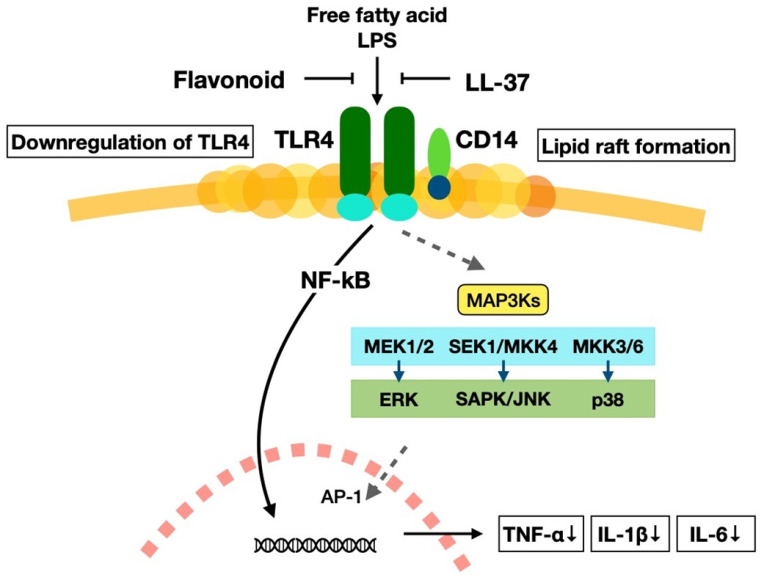
The mechanism of the inactivation of proinflammatory cytokine production pathways by the flavonoids and LL-37 through raft formation. Free fatty acid and lipopolysaccharide (LPS) activate nuclear factor-κ B (NF-κB) through toll-like receptor 4 (TLR4) pathway: After macrophage receptors are stimulated with these ligands, the components of the cell membrane form lipid rafts to transmit signals into the cytoplasm and activate transcription factor NF-κB. Several signalling pathways exist under the downstream of TLR4, like mitogen-activated protein kinase kinase kinase 3 (MAP3Ks), which is associated with cell proliferation and cell death and is important for the variety of physiological functions. Activated NF-κB is associated with the transcription of proinflammatory cytokine genes; however, the flavonoid quercetin and antimicrobial peptide LL-37 inhibit the activation of NF-κB. This mechanism stops the production of proinflammatory cytokines TNF-α, IL-1β, and IL-6. These cytokines decrease in the body and stop the onset of chronic inflammation.

**Figure 4 ijms-25-02125-f004:**
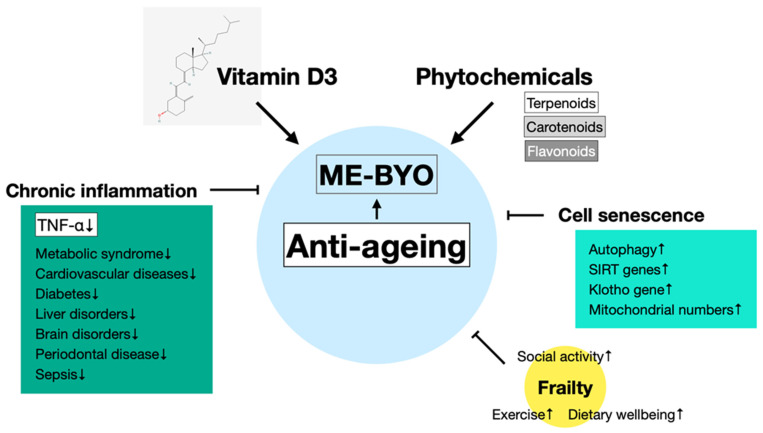
Schematic representation of anti-ageing effects of vitamin D3 and phytochemicals. Vitamin D3 and phytochemicals suppress not only frailty, a bone-related illness associated with ageing, but also senescence of cells and chronic inflammation, and reduces the risk of related diseases. These effects manifest anti-ageing effects through the prevention of “ME-BYO” (a non-disease condition) and participate in healthy longevity.

**Table 1 ijms-25-02125-t001:** Variety and effects of flavonoid phytochemicals.

Flavonoids	Effects	Ref.
Quercetin, naringin, nobiletin	Anti-inflammatory effects in metabolic syndrome prevention	[66]
Flavan-3-ols, flavon-3-ols, anthocyanidins	Anti-inflammatory effects in intestinal disorders	[67]
Quercetin, hesperidin, nobiletin, rutin	Anti-inflammatory effects for healthy longevity and prevention of “ME-BYO”	[68]
Catechins	Cellular antioxidant properties	[73]
Theaflavin	Health-related effects and chemistry	[74]
Anthocyanin	Pharmaceutical and nutraceutical effects other than use as colour pigments	[75]
Hesperidin	Hesperidin in obesity treatment	[76]
Isoflavone	Therapeutic effects from oestrogen activity	[77]
Sesamin	Effects of sesamin in angiogenic processes	[78]
Flavonoids	Anti-inflammation effects of flavonoids	[88]
Quercetin	Anti-ageing effects	[89]
Quercetin, fisetin	Effects of flavonoids in cancer chemotherapy	[90]
Quercetin, luteolin	Suppression of TNF-α production of flavonoids in cellular process	[91]
Hesperidin	Suppression of endotoxic shock	[92]

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
