# Peer review of "The Potential Use of Vitamin D3 and Phytochemicals for Their Anti-Ageing Effects"

_ijms, 2024, doi:10.3390/ijms25042125_

Round 1

Reviewer 1 Report

Comments and Suggestions for Authors

In this review, the effects of vitamin D in suppressing inflammatory responses by interacting with phytochemicals and balancing the immune response and link with anti-aging are discussed.
1. The manuscript needs revision in terms of English writing and grammar.
2. A section should be considered for studies that have been conducted with the simultaneous use of vitamin D with phytochemicals.
3. The limitation of the use of some phytochemicals is their low solubility, which reduces their bioavailability. The methods of solving this problem can be discussed, for example using nanotechnology
4. Can phytochemicals mimic the action of vitamin D by binding to the VDR receptor? It is suggested to write about this. For example, research has been done on the binding of curcumin to VDR.
5. It is suggested to write about future perspectives in the discussion section.

Comments on the Quality of English Language

The manuscript needs revision in terms of English writing and grammar.

Author Response

RESPONSES TO REVIEWERS

Response to Reviewer 1

Comments and Suggestions for Authors
In this review, the effects of vitamin D in suppressing inflammatory responses by interacting with phytochemicals and balancing the immune response and link with anti-aging are discussed.

Thank you very much for your comments. In accordance with the comments from reviewers, we have revised our manuscript.

1. The manuscript needs revision in terms of English writing and grammar.

Response: we ordered English editing service of the publisher and acquired the certificate.

2. A section should be considered for studies that have been conducted with the simultaneous use of vitamin D with phytochemicals.
Response: To make the passages more clearly, in “(line-399) 5. Synergy between vitamin D3 and phytochemicals”, we have changed the passages and added the references to explain the relationship between the vitamin D3 and phytochemicals.

The name of the section “5. Synergy between vitamin D and phytochemicals” has been changed to “(line-399) 5. Synergy between vitamin D3 and phytochemicals”.

The passages “Activated vitamin Binds to the vitamin D receptor and stop the expression of TNF-α gene by binding to the promoter region. Figure 3 shows the mechanism of the inhibition of TNF-α gene expression through the raft formation by flavonoids.” has been changed to “(line-411) Figure 3 shows the mechanism of the inhibition of TNF-α gene (TNF) expression via raft formation by flavonoids. In addition, activated vitamin D3 (1α,25-(OH)2-D3) binds to the VDR and stops the expression of the TNF gene by binding to the promoter region as shown in Figure 1. These interaction between the vitamin D3 and phytochemicals inhibits the production of TNF-α and stop the progression of ageing through the prevention of chronic inflammation.”.

The name of the section “5.1 Bone metabolism” has been changed to “(line-426) 5.1 Vitamin D3 and phytochemicals in bone metabolism”.

New reference has been added: “(line-431) TGR5-expressing intestinal L cells and pancreatic α cells increase the expression of GLP-1, and pancreatic β cells enhance insulin secretion, preventing the onset of diabetes [95].”.

The passage “In addition, oleanolic acid suppresses inflammation by suppressing the production of IL-12 and TNF-α from dendritic cells, cytokine production in Kupffer cells, and TNF-α, IL-6, and IL-1β production in macrophages.” has been changed and new reference were added to “(line-435) In addition, oleanolic acid suppresses inflammation by suppressing the production of IL-12 and TNF-α in dendritic cells; cytokine

page2image46700640

2

production in Kupffer cells; and proinflammatory cytokines TNF-α, IL-1β, and IL-6 production in macrophages [96,97]. ”.

New reference has been added: “(line-441) Research into the effects of the combined administration of vitamin D3 and tea catechins has been conducted for the inhibition of bone metabolism by osteoclasts [98].”.

New reference has been added: “(line-443) Citrus unshiu mandarin oranges contain large amounts of the carotenoid β-cryptoxanthin, and research by the National Agriculture and Food Research Organisation (NARO) has analysed their ability to prevent the onset of osteoporosis [99]. ”.

The name of the subsection “5.2 Prevention of anti-ageing” has been changed to “(line-447) 5.2 Vitamin D3 and phytochemicals in the prevention of ageing”.

The new passage has been added “(line-487) Therefore, vitamin D3 and phytochemicals have a synergistic effect on anti-ageing by activating anti-ageing associated genes.”.

The name of the subsection “5.3 Suppression of chronic inflammation” has been changed to “(line-489) 5.3 Vitamin D3 and phytochemicals in the suppression of chronic inflammation”.

New reference has been added: “(line-496) The Maillard reaction, a chemical glycation process between the sugar and amino groups of proteins produces glycated proteins instead of glycosylation [117].”.

New reference has been added: “(line-500) Flavonoids suppress the TNF-α production induced by AGEs and therefore suppress ageing [118].”.

New reference has been added: “(line-501) AGEs induce inflammation in the brain, and the suppression of glucose uptake by this inflammation induces both cell death and diseases such as Alzheimer’s disease [119].”.

New reference has been added: “(line-504) The intake of vitamin D3 and flavonoids improves quality of life (QOL), suppresses ageing, and is associated with a long and healthy life [120].”.

The passage “In the Mediterranean diet, red meat consumption is about once a week however, recently influenced by the United States, consumption of red meat is recommended in Japan” has been changed to “(line-522) In the Mediterranean diet, which is known as healthy diet worldwide, red meat is consumed about once a week; however, recently, influenced by the United States, the consumption of red meat was recommended in Japan.”.

The new passage has been added “(line-532) It is well known that vitamin D3 participate in the prevention of osteoporosis. The upregulation of TNF-α production by AGEs is also suppressed by vitamin D3 and phytochemicals. Therefore, this interaction is effective in anti-ageing by suppressing chronic inflammation.”.

New reference has been added: “(line-552) The Japanese diet is being recognised for its association with longevity. The diet that leads to a healthy and long life is the diet consumed around 1975 in Japan [135].”.

New reference has been added: “(line-563) It is also known to enhance GSL-1-mediated effects [136].”. 3

The passages “In addition to vitamin D and phytochemicals from vegetables and fruits, ω-3 fatty acids EPA and DHA are important in the suppression of chronic inflammation.” has been changed to “(line-571) As this review has shown above, the supplementation of vitamin D3 and phytochemicals from vegetables and fruits are effective strategies for anti-ageing. In addition, the ω-3 fatty acids EPA and DHA have important roles in the suppression of chronic inflammation.”.

3. The limitation of the use of some phytochemicals is their low solubility, which reduces their bioavailability. The methods of solving this problem can be discussed, for example using nanotechnology
Response: We mentioned about the use of nanotechnology to improve the bioavailability of phytochemicals in our body. In addition, we mentioned about how phytochemicals moves from intestine to bloodstream before the manifestation of the effects.

The new subsection has been added:
“(line-375) 4.4. Attempts to improve the bioavailability and activation of phytochemicals in the body”.

“(line-376) Low bioavailability of the phytochemicals is well known problem. Generally, phytochemicals have low solubility to the water, making them difficult to reach target organs and cells. Therefore, the application of nanotechnology has been attractive choice from the perspective of drug delivery system (DDS) for phytochemicals to reach the targets especially in cancer therapy. Efforts are being made to deliver these chemicals into specific organs or cells by creating capsules using nanotechnology. This attempt is being conducted with high expectancy [83,84].”.

“(line-383) Another aspect is the chemical modification of phytochemicals. Compounds like hesperidin and quercetin are insoluble in the water by themselves however, they become water-soluble after the glucosylation and reach to the intestines [85]. Thus, they are broken down by the enzymes from gut microbiota to hesperitin and quercetin, and are able to passing through the intestinal cell wall. After the absorption, quercetin conjugates with glucronide by the other enzyme and absorbed into the bloodstream [86]. In atherosclerosis, quercetin-glucronide is taken up by the foam cells in the blood vessels and become active form after the removal of glucronide by the enzyme secreted from the cells. Activated quercetin manifest the effects and suppress the formation of thrombosis [87].”.

4. Can phytochemicals mimic the action of vitamin D by binding to the VDR receptor? It is suggested to write about this. For example, research has been done on the binding of curcumin to VDR.
Response: We mentioned that curcumin has physiological role through binding to the VDR, as you suggested. Furthermore, we described recent finding of flavonoid quercetin that can interact with VDR and prevent breast cancer and fibrosis.

The new subsection has been added:
“(line-392) 4.5. Consideration of the affinity of phytochemicals to the VDR”.

“(line-393) Other researchers have suggested that phytochemicals directly bind to VDR. For instance, curcumin, a phytochemical contained in turmeric binds to the VDR as a ligand and exhibit physiological roles [88,89,90]. Moreover, the combination of vitamin D3 and curcumin has been reported to prevent the ageing of the brain and maintenance of healthy conditions [91]. According to the latest research, flavonoid quercetin also interacts with VDR and prevent breast cancer and fibrosis, and participants in anti-ageing [92].”.

4

5. It is suggested to write about future perspectives in the discussion section.

Response: We added the new passage to “(line-588) 6. Conclusions”.
The new passage has been added to “(line-599) Recently, attention to the health longevity has been increasing to an unprecedented level. Hence, it is expected that research on the prevention of lifestyle-related diseases and age-related diseases caused by chronic inflammation through the intake of vitamin D3 and phytochemicals must be increasing in the future.”.

Comments on the Quality of English Language
The manuscript needs revision in terms of English writing and grammar.
Response: we ordered English editing service of the publisher and acquired the certificate.

Reviewer 2 Report

Comments and Suggestions for Authors

The manuscript explored an interesting topic. It was well-organized and supported by appropriate references; however, some modifications are suggested.

The rationale of the study should be explained better. The necessity of such research should be more highlighted.

The abstract should be modified and provide more information with better structure. 

Some acronyms do not have the corresponding clarification the first time they appear in the text and abstract, such as Tumor necrosis factor-alpha (TNF-α) ;Coronavirus disease 2019 (COVID-19) ...

A brief search strategy should be provided in method section.

It is recommended to replace instances of "we" with alternative phrases such as "current study," "this study," and "present study" in the manuscript.

      Comments on the Quality of English Language

Moderate proofreading and paraphrasing are required.

Author Response

Response to Reviewers

Comments and Suggestions for Authors
The manuscript explored an interesting topic. It was well-organized and supported by appropriate references; however, some modifications are suggested.

Thank you very much for your helpful advice. We have revised our manuscript in accordance with your comment.

1. The rationale of the study should be explained better. The necessity of such research should be more highlighted.
Response: We have revised our manuscript in accordance with the point out from the reviewers. Regarding to “vitamin D”, we mainly focused on vitamin D3 rather than vitamin D2. In addition, the section “4. phytochemicals” introduced subsections and added new information regarding bioavailability and influence on VDR. Furthermore, we describe the necessity of the research in “1. introduction” and future prospects in “6. Conclusion”.

2. The abstract should be modified and provide more information with better structure.
Response: we added more information and example to provide better structure in “Abstract”.
“(line-15) Abstract: Unlike other vitamins, vitamin D3 is synthesised in cells in the body. Vitamin D3 has been known as a bone-related hormone. Recently, however, it has been considered as a vitamin involved in immunity. Vitamin D3 deficiency influences the onset of a variety of diseases. Vitamin D3 regulates the production of proinflammatory cytokines such as TNF-α through the binding to vitamin D receptors (VDR) in immune cells. Since blood levels of vitamin D3 (25-OH-D3) were low even in coronavirus disease 2019 (COVID-19) infection, there has been growing interest in the importance of vitamin D3. On the other hand, phytochemicals are compounds derived from plants with over 7,000 varieties and they have various biological activities. They mainly have health-promoting effects and are classified as terpenoids, carotenoids, flavonoids, etc. Flavonoids are known as the anti-inflammatory compounds that control TNF-α production. Chronic inflammation is induced by the continuous production of TNF-α. Chronic inflammation is the fundamental cause of diseases like obesity, dyslipidaemia, diabetes, heart and brain diseases, autoimmune diseases, Alzheimer’s disease, and cancer. In addition, ageing is induced by chronic inflammation. This review discusses the effects of vitamin D3 in the suppression of inflammatory responses as well as its interaction with phytochemicals, how it balances the immune response, and its link to anti-ageing effects. In addition, vitamin D3 and phytochemicals are synergistically contributed to anti-ageing by working with ageing-related genes. Furthermore, prevention of ageing processes through the chronic inflammation requires the maintenance of healthy gut microbiota, which is related with daily dietary habits. In this regard, supplementation of vitamin D3 and phytochemicals plays an important role. Recently, the association of the prevention of the non-disease condition called “ME-BYO” with the maintenance of a healthy condition was revealed, and the anti-ageing effect discussed here is important for healthy longevity.”.

 3. Some acronyms do not have the corresponding clarification the first time they appear in the text and abstract, such as Tumor necrosis factor-alpha (TNF-α) ;Coronavirus disease 2019 (COVID-19) ... Response: we added the meaning of each acronyms in the manuscript.
Tumor necrosis factor-α (TNF-α)
Coronavirus disease 2019 (COVID-18)
Fibroblast growth factor 23 (FGF23)
Cytochrome P450 family 24 subfamily A member 1 (CYP24A1) Low-density lipoprotein (LDL)

4. A brief search strategy should be provided in method section.
Response: we aded detailed description in the section:
“(line-94) 2. Method”.
“(line-95) This section briefly describes the method used to search for references for this review. This is a narrative review: information was collected using PubMed search with complexed keywords including “vitamin D3”, “phytochemicals”, “anti-ageing”, “gut microbiota”, “chronic inflammation” and “immunity”. Furthermore, filtering functions have been applied such as searching only article type “review” and publication data limited within recent 5 years. References cited were written in English and peer reviewed papers. In addition to academic papers, this article also widely referred to press releases from universities and research institutes, as well as other scientific featured articles from the newspapers and scientific information webpages and their reference papers. The current research referred to the latest references, except for references considered to be important, and over 40% of the references were limited to the last five years.”.

5. It is recommended to replace instances of "we" with alternative phrases such as "current study," "this study," and "present study" in the manuscript.
Response: we changed almost all the passages including “we” to alternative phrases as shown below.
1. The part of passage “In this review, we discuss... “ has been changed to “(line-26) This review discusses...”.
2. The part of passage “Briefly, we describe...” has been changed to “(line-95) This section briefly describes...”.
3. The part of passage “we also referred to...” has been changed to “(line-101) this article also widely referred
to...“.
4. The part of passage “We need to recognise...” has been changed to “(line-285) It is important to
recognise...”.
5. The part of the passage “Here, we describe...” has been changed to “(line-333) Here, this review describes...“.
6. The part of the passage “We have already mentioned regarding to the phytoestrogens...” has been changed to “(line-440) The effects of phytoestrogens...“.
7. The part of the passage “As we mentioned above,...” has been changed to “(line-480) As mentioned above,...“.

Comments on the Quality of English Language
Moderate proofreading and paraphrasing are required.
Response: we ordered English editing service of the publisher and acquired the certificate.

Reviewer 3 Report

Comments and Suggestions for Authors

Major comments:

1. There should be more and appropriate references.

2. Please be consistent in the nomenclature of vitamin D compounds. Moreover, indicate which compound is meant when you use the term "vitamin D".

3. The Method section is far too short.

4. The four of the manuscript is unclear. Moreover, it lacks new insight.

Minor comments:

1. All abbreviations should be defined at first time use and then consistently applied. Thus concerns also the Abstract.

2. Gene name abbreviations should be in italic. Moreover, please use latest gene name nomenclature (e.g. TNF, not TNF-alpha)

Comments on the Quality of English Language

English language needs significant revision.

Author Response

Response to Reviewers

Comments and Suggestions for Authors

Thank you very much for your review. We have made a revision of our manuscript as shown below.

Major comments:
1. There should be more and appropriate references.
Response: We added references to the place in the manuscript where only a few information from reference were provided.

27. Vernia F.; Valvano M.; Longo S.; Cesaro N.; Viscido A.; Latella G. Vitamin D in Inflammatory Bowel Diseases. Mechanisms of Action and Therapeutic Implications. Nutrients 2022, 14, 269.
32. Shenoy S. Gut microbiome, Vitamin D, ACE2 interactions are critical factors in immune-senescence and inflammaging: key for vaccine response and severity of COVID-19 infection. Inflamm. Res. 2022, 71, 13-26.
33. Dimitrov V.; White JH. Vitamin D signaling in intestinal innate immunity and homeostasis. Mol. Cell Endocrinol. 2017, 453, 68-78.
36. Krutzik SR.; Hewison M.; Liu PT.; Robles JA.; Stenger S.; Adams JS.; Modlin RL. IL-15 links TLR2/1- induced macrophage differentiation to the vitamin D-dependent antimicrobial pathway. J. Immunol. 2008, 181, 7115-7120.
37. Takahashi K.; Nakayama Y.; Horiuchi H.; Ohta T.; Komoriya K.; Ohmori H.; Kamimura T. Human neutrophils express messenger RNA of vitamin D receptor and respond to 1alpha,25-dihydroxyvitamin D3. Immunopharmacol. Immunotoxicol. 2002, 24, 335-347.
38. Li W.; Che X.; Chen X.; Zhou M.; Luo X.; Liu T. Study of calcitriol anti-aging effects on human natural killer cells in vitro. Bioengineered 2021, 12, 6844-6854.
46. Zhang R.; Wang M.; Wang M.; Zhang L.; Ding Y.; Tang Z.; Fu Z.; Fan H.; Zhang W.; Wang J. Vitamin D Level and Vitamin D Receptor Genetic Variation Were Involved in the Risk of Non-Alcoholic Fatty Liver Disease: A Case-Control Study. Front. Endocrinol. (Lausanne) 2021, 12, 648844.
47. Chen J.; Zhang J.; Li J.; Qin R.; Lu N.; Goltzman D.; Miao D.; Yang R. 1,25-Dihydroxyvitamin D Deficiency Accelerates Aging-related Osteoarthritis via Downregulation of Sirt1 in Mice. Int. J. Biol. Sci. 2023, 19, 610-624.
62. Santa K. Grape Phytochemicals and Vitamin D in the Alleviation of Lung Disorders. Endocr Metab Immune Disord Drug Targets. 2022, 22, 1276-1292.
63. Santa K. Healthy Diet, Grape Phytochemicals, and Vitamin D: Preventing Chronic Inflammation and Keeping Good Microbiota. Endocr Metab Immune Disord Drug Targets. 2023, 23, 777-800.
65. Marko M.; Pawliczak R. Resveratrol and Its Derivatives in Inflammatory Skin Disorders-Atopic Dermatitis and Psoriasis: A Review. Antioxidants (Basel) 2023, 12, 1954.
77. Chávez-Talavera O.; Tailleux A.; Lefebvre P.; Staels B. Bile Acid Control of Metabolism and Inflammation in Obesity, Type 2 Diabetes, Dyslipidemia, and Nonalcoholic Fatty Liver Disease. Gastroenterology 2017, 152, 1679-1694.e3.
83. Xie J.; Yang Z.; Zhou C.; Zhu J.; Lee RJ.; Teng L. Nanotechnology for the delivery of phytochemicals in cancer therapy. Biotechnol. Adv. 2016, 34, 343-353.
84. Kim B.; Park JE.; Im E.; Cho Y.; Lee J.; Lee HJ.; Sim DY.; Park WY.; Shim BS.; Kim SH. Recent Advances in Nanotechnology with Nano-Phytochemicals: Molecular Mechanisms and Clinical Implications in Cancer Progression. Int. J. Mol .Sci. 2021, 22, 3571.
85. Pyrzynska K. Hesperidin: A Review on Extraction Methods, Stability and Biological Activities. Nutrients 2022, 14, 2387.
86. Amaretti A.; Raimondi S.; Leonardi A.; Quartieri A.; Rossi M. Hydrolysis of the rutinose-conjugates flavonoids rutin and hesperidin by the gut microbiota and bifidobacteria. Nutrients 2015, 7, 2788-2800.
87. Kawai Y.; Nishikawa T.; Shiba Y.; Saito S.; Murota K.; Shibata N.; Kobayashi M.; Kanayama M.; Uchida
K.; Terao J. Macrophage as a target of quercetin glucuronides in human atherosclerotic arteries: implication in the anti-atherosclerotic mechanism of dietary flavonoids. J. Biol. Chem. 2008, 283, 9424-9434.
88. Gagliardi S.; Franco V.; Sorrentino S.; Zucca S.; Pandini C.; Rota P.; Bernuzzi S.; Costa A.; Sinforiani E.; Pansarasa O.; Cashman JR.; Cereda C. Curcumin and Novel Synthetic Analogs in Cell-Based Studies of Alzheimer's Disease. Front. Pharmacol. 2018, 9, 1404.
89. Bartik L.; Whitfield GK.; Kaczmarska M.; Lowmiller CL.; Moffet EW.; Furmick JK.; Hernandez Z.; Haussler CA.; Haussler MR.; Jurutka PW. Curcumin: a novel nutritionally derived ligand of the vitamin D receptor with implications for colon cancer chemoprevention. J. Nutr. Biochem. 2010, 21, 1153-1161.
90. Haussler MR.; Whitfield GK.; Kaneko I.; Haussler CA.; Hsieh D.; Hsieh JC.; Jurutka PW. Molecular mechanisms of vitamin D action. Calcif. Tissue Int. 2013, 92, 77-98.
91. Farghali M.; Ruga S.; Morsanuto V.; Uberti F. Can Brain Health Be Supported by Vitamin D-Based Supplements? A Critical Review. Brain Sci. 2020, 10, 660.
92. Sannappa Gowda NG.; Shiragannavar VD.; Puttahanumantharayappa LD.; Shivakumar AT.; Dallavalasa S.; Basavaraju CG.; Bhat SS.; Prasad SK.; Vamadevaiah RM.; Madhunapantula SV.; Santhekadur PK. Quercetin activates vitamin D receptor and ameliorates breast cancer induced hepatic inflammation and fibrosis. Front. Nutr. 2023, 10, 1158633.
95. Wang Q.; Lin H.; Shen C.; Zhang M.; Wang X.; Yuan M.; Yuan M.; Jia S.; Cao Z.; Wu C.; Chen B.; Gao A.; Bi Y.; Ning G.; Wang W.; Wang J.; Liu R. Gut microbiota regulates postprandial GLP-1 response via ileal bile acid-TGR5 signaling. Gut. Microbes. 2023, 15, 2274124.
96. Wang J.; Wang J.; Hong W.; Zhang L.; Song L.; Shi Q.; Shao Y.; Hao G.; Fang C.; Qiu Y.; Yang L.; Yang Z.; Wang J.; Cao J.; Yang B.; He Q.; Weng Q. Optineurin modulates the maturation of dendritic cells to regulate autoimmunity through JAK2-STAT3 signaling. Nat. Commun. 2021, 12, 6198.
97. Li W.; Zeng H.; Xu M.; Huang C.; Tao L.; Li J.; Zhang T.; Chen H.; Xia J.; Li C.; Li X. Oleanolic Acid Improves Obesity-Related Inflammation and Insulin Resistance by Regulating Macrophages Activation. Front. Pharmacol. 2021, 12, 697483.
98. Welch AA.; Hardcastle AC. The effects of flavonoids on bone. Curr. Osteoporos. Rep. 2014, 12, 205-210.
99. Maeda-Yamamoto M.; Ohtani T. Development of functional agricultural products utilizing the new
health claim labeling system in Japan. Biosci. Biotechnol. Biochem. 2018, 82, 554-563.
117. Gill V.; Kumar V.; Singh K.; Kumar A.; Kim JJ. Advanced Glycation End Products (AGEs) May Be a
Striking Link Between Modern Diet and Health. Biomolecules 2019, 9, 888.
118. Filošević Vujnović A.; Jović K.; Pištan E, Andretić Waldowski R. Influence of Dopamine on Fluorescent
Advanced Glycation End Products Formation Using Drosophila melanogaster. Biomolecules 2021, 11, 453.
119. Münch G.; Kuhla B.; Lüth HJ.; Arendt T, Robinson SR. Anti-AGEing defences against Alzheimer's
disease. Biochem. Soc. Trans. 2003, 31, 1397-1399.
120. Szczechowiak K.; Diniz BS.; Leszek J. Diet and Alzheimer's dementia - Nutritional approach to modulate
inflammation. Pharmacol. Biochem. Behav. 2019, 184, 172743.
135. Kushida M.; Sugawara S.; Asano M.; Yamamoto K.; Fukuda S.; Tsuduki T. Effects of the 1975 Japanese
diet on the gut microbiota in younger adults. J. Nutr. Biochem. 2019, 64, 121-127.
136. Johmura Y.; Yamanaka T.; Omori S.; Wang TW.; Sugiura Y.; Matsumoto M.; Suzuki N.; Kumamoto S.;
Yamaguchi K.; Hatakeyama S.; Takami T.; Yamaguchi R.; Shimizu E.; Ikeda K.; Okahashi N.; Mikawa R.; Suematsu M.; Arita M.; Sugimoto M.; Nakayama KI.; Furukawa Y.; Imoto S.; Nakanishi M. Senolysis by glutaminolysis inhibition ameliorates various age-associated disorders. Science 2021, 371, 265-270.

2. Please be consistent in the nomenclature of vitamin D compounds. Moreover, indicate which compound is meant when you use the term "vitamin D”.
Response: Even though vitamin D2 and vitamin D3 are mainly considered as vitamin D, we focused on vitamin D3.
Title has been changed to “(line-2) The potential use of vitamin D3 and phytochemicals for their anti-ageing effects”.
The synthesis and metabolic pathway of vitamin D3 has been shown in “(line-121) Figure 1”, which including the synthesis of vitamin D3 from cholesterol to stop the production of TNF-α.
In this manuscript, all of the blood (or serum) vitamin D3 has been described as “25-OH-D3”and activated form of vitamin D3 has been described as “1α,25-(OH)2-D3”.

3. The Method section is far too short.
Response: we aded detailed description in the section:
“(line-94) 2. Method”.
“(line-95) This section briefly describes the method used to search for references for this review. This is a narrative review: information was collected using PubMed search with complexed keywords including “vitamin D3”, “phytochemicals”, “anti-ageing”, “gut microbiota”, “chronic inflammation” and “immunity”. Furthermore, filtering functions have been applied such as searching only article type “review” and publication data limited within recent 5 years. References cited were written in English and peer reviewed papers. In addition to academic papers, this article also widely referred to press releases from universities and research institutes, as well as other scientific featured articles from the newspapers and scientific information webpages and their reference papers. The current research referred to the latest references, except for references considered to be important, and over 40% of the references were limited to the last five years.”.

4. The four of the manuscript is unclear. Moreover, it lacks new insight.
Response: At first, we explained that this review mention the phytochemicals mainly contained in grapes. Then, we divided the paragraph to several subsections including “(line-341) 4.1. Flavonoid polyphenols”, “(line-362) 4.2. Terpenoids” and “(line-364) 4.3. Carotenoids”.
In addition, as a new insight, we added new subsections “(line-375) 4.4. Attempts to improve the bioavailability and activation of phytochemicals in the body” and discussed regarding to low availability of phytochemicals and shown the example of the solutions. Furthermore, another new subsection has been added “(line-392) 4.5. Consideration of the affinity of phytochemicals to the VDR” and discussed the affinity of several phytochemicals to the VDR in accordance to the opinion from reviewers.

Changes in “(line-328) 4. Phytochemicals”.
The passage “Terpenoids, carotenoids, flavonoids and even β-glucans are included in phytochemicals a variety of plant-drives chemicals.” has been changed to “(line-329) As we have reported previously [59-63], terpenoids, carotenoids, flavonoids, and even β-glucans are included in phytochemicals, a variety of plant- derived chemicals.”.
The new passage has been added “(line-334) Especially, current review mainly focusses on phytochemicals contained in grapes that are particularly noteworthy. The attractiveness of grape phytochemicals started with the red wine polyphony resveratrol, which gained the attention due to the “French Paradox” — where the mortality rate from cardiovascular disease is one third of the U.S. In addition, since the Mediterranean diet that is considered as healthy around the world contains lots of phytochemicals, they became famous [65]. ”.
The new subsection as been added “(line-341) 4.1. Flavonoid polyphenols”. The new subsection as been added “(line-352) 4.2. Terpenoids”.
The passage “Terpenoids are the chemicals with triterpene skeleton structure including oleanolic acid, ursolic acid, and sugar attached saponin and oleanolic acid is a component of the white powder called bloom on the surface of grapes working as an agonist for the bile acid receptor TGR5 [64,65].” has been changed to “(line-353) Terpenoids are chemicals with a triterpene skeleton structure, including oleanolic acid, ursolic acid, and saponin (sugar attached), and oleanolic acid is a component of the white powder called bloom on the surface of grapes, which works as an agonist of the bile acid receptor TGR5 on the cell surface [75,76].”.
The new passage with new reference has been added to “(line-356) In addition, since oleanolic acid works as an agonist of nuclear receptor FXR, it activates genes and transcriptional networks associated with the metabolism of sugar and lipids, energy consumption, and inflammation [77].”.
The passage “” has been changed to “(line-359) Furthermore, oleanolic acid plays an important role in vitamin D3 metabolism, enhancing CYP27B1 for the generation of the active form of vitamin D3 (1α,25- (OH)2-D3) and suppressing CYP24A1 for the decomposition of the 1α,25-(OH)2-D3.”.
The new subsection as been added “(line-364) 4.3. Carotenoids”.
The new subsection has been added:
“(line-375) 4.4. Attempts to improve the bioavailability and activation of phytochemicals in the body”.
“(line-376) Low bioavailability of the phytochemicals is well known problem. Generally, phytochemicals have low solubility to the water, making them difficult to reach target organs and cells. Therefore, the application of nanotechnology has been attractive choice from the perspective of drug delivery system (DDS) for phytochemicals to reach the targets especially in cancer therapy. Efforts are being made to deliver these chemicals into specific organs or cells by creating capsules using nanotechnology. This attempt is being conducted with high expectancy [83,84].”.
“(line-383) Another aspect is the chemical modification of phytochemicals. Compounds like hesperidin and quercetin are insoluble in the water by themselves however, they become water-soluble after the glucosylation and reach to the intestines [85]. Thus, they are broken down by the enzymes from gut microbiota to hesperitin and quercetin, and are able to passing through the intestinal cell wall. After the absorption, quercetin conjugates with glucronide by the other enzyme and absorbed into the bloodstream [86]. In atherosclerosis, quercetin-glucronide is taken up by the foam cells in the blood vessels and become active form after the removal of glucronide by the enzyme secreted from the cells. Activated quercetin manifest the effects and suppress the formation of thrombosis [87].”.
The new subsection has been added:
“(line-392) 4.5. Consideration of the affinity of phytochemicals to the VDR”.
“(line-393) Other researchers have suggested that phytochemicals directly bind to VDR. For instance, curcumin, a phytochemical contained in turmeric binds to the VDR as a ligand and exhibit physiological roles [88,89,90]. Moreover, the combination of vitamin D3 and curcumin has been reported to prevent the ageing of the brain and maintenance of healthy conditions [91]. According to the latest research, flavonoid quercetin also interacts with VDR and prevent breast cancer and fibrosis, and participants in anti-ageing [92].”.

Minor comments:
1. All abbreviations should be defined at first time use and then consistently applied. Thus concerns also the Abstract.
Response: we added the meaning of each abbreviations in the manuscript.
Tumor necrosis factor-α (TNF-α)
Coronavirus disease 2019 (COVID-18)
Fibroblast growth factor 23 (FGF23),
Cytochrome P450 family 24 subfamily A member 1 (CYP24A1)
Low-density lipoprotein (LDL)

2. Gene name abbreviations should be in italic. Moreover, please use latest gene name nomenclature (e.g. TNF, not TNF-alpha)
Response: We check the database in “National Center for Biotechnology Information” and corrected all the name of Genes in the manuscript including “TNF, CYP27B1, CAMP, DEFB4A”.
Comments on the Quality of English Language
English language needs significant revision.
Response: we ordered English editing service of the publisher and acquired the certificate.

Round 2

Reviewer 1 Report

Comments and Suggestions for Authors

Corrections have been made and the manuscript is acceptable.

Author Response

Comments and Suggestions for Authors
Corrections have been made and the manuscript is acceptable.

Response: Thank you very much for your review. In accordance with the other reviewer’s comments (reviewer 3) we have extensively checked and corrected our manuscript. All the changes has been shown in the attached PDF file.

We sincerely hope our manuscript will be accepted and published in the near future in International Journal of Molecular Sciences.

Reviewer 2 Report

Comments and Suggestions for Authors

The authors tried to address the main comments. The manuscript has potential to be published.

Comments on the Quality of English Language

Moderate proofreading and paraphrasing are required.

Author Response

Comments and Suggestions for Authors

The authors tried to address the main comments. The manuscript has potential to be published.

Thank you very much for your review and comment. In accordance with the other reviewer’s comment we have intensively checked and revised our manuscript as shown below. (Response to reviewer 3)

Comments on the Quality of English Language

Moderate proofreading and paraphrasing are required.

Response: As shown below, English written manuscript has been intensively checked and changed. We attempted to paraphrase the passages in the manuscript.

We sincerely hope the changed manuscript is suitable to be published in International Journal of Molecular Sciences.

Response to Reviewer (3) has been shown below.

1. The resolution of the figures is very insufficient, please improve.

Response: Figures in the manuscript have been exported to PDF format from software used (Keynote Mac) and sent directly to the Editorial Office. We did not use Adobe Illustrator but figures in PDF format should be no problems.

2. Please have the nomenclature of genes also correct in figures (italic and latest nomenclature)

Response: In “(line-919) Figure 1” and “(line-942) Figure 3”, the name of cytokine proteins (TNF-α, IL-1β, IL-6) have been changed from italic to normal font. Figures were changed to show these proteins exist outside the nucleus.

Genes in the manuscript have been shown in the italic font.

TNF

CYP27B1

Klotho (KL)

VDR

CD14

CAMP

β-defensin 2 (DEFB4A)

SIRT1

SIRT7

3. Please have a clear understanding of the vitamin D nomenclature and revise.

Response: As the explanation of vitamin D3 has not been precise, passages in “1. Introduction” and “3. Vitamin D3” has been exchanged. The process of the generation of active form of vitamin D3 from provitamin D3, and process of inactivation have been summarised in the first paragraph of “3. vitamin D3”.

“(line-39) 1. Introduction”

The passage “The vitamin D3 precursor provitamin D3 (7-dehydrocholesterol) is converted to previtamin D3 and then to vitamin D3.” has been changed to “(line-55) Vitamin D3 is produced from cholesterol in skin cells after exposure to ultraviolet rays from sunlight.”.

“(line-109) 3. Vitamin D3”

The passage “Vitamin D3 is produced from cholesterol in skin cells after exposure to ultraviolet rays from sunlight.” has been changed to “(line-114) The vitamin D3 precursor provitamin D3 (7-dehydrocholesterol) is converted to previtamin D3 and then to vitamin D3.”.

The passage “Vitamin D3 circulates in the blood stream after binding to vitamin D-binding protein (VDBP) and is converted to 25-OH-D3 (calcidiol) by a liver enzyme.” has been changed to “(line-115) Vitamin D3 circulates in the blood stream after binding to vitamin D binding protein (VDBP) and is converted to 25-OH-D3 (calcidiol) by liver enzymes such as cytochrome P450 2R1 (CYP2R1).”.

The passage “Enzymes in the kidneys and immune cells convert 25-OH-D3 to the active form, 1α,25-(OH)2-D3 (calcitriol), which binds to the vitamin D receptor (VDR) in the cytoplasm and translocate into the nucleus [12,13,14].” has been change to “(line-117) An enzyme CYP27B1 in the kidneys and immune cells convert 25-OH-D3 to the active form, 1α,25-(OH)2-D3 (calcitriol), which binds to the vitamin D receptor (VDR) in the cytoplasm and translocate into the nucleus [12,13,14].”.

The passage has been added “(line-125) Activated form of vitamin D3 (1α,25-(OH)2-D3) inactivated to 24(R),25-(OH)2-D3 (24,25-dihydroxycholecalciferol) within a few hours by the enzyme CYP24A1.”.

The passage with new reference (1) has been added “(line-157) Vitamin D3 also mitigates the risk of bone density loss induced by cadmium poisoning [27].”.

“(line-159) 3.1 Vitamin D3 and infectious diseases ”

The new reference (2) has been added “(line-166) In this context, vitamin D3 stops the production of TNF-α and suppresses its responses to avoid inducing excess and chronic inflammation [29].”.

The passage “The onset of seasonal influenza and COVID-19 is associated with serum vitamin D3 (25-OH-D3) levels.” has been changed to “(line-169) The onset of seasonal flu is associated with serum 25-OH-D3 levels.”.

The passage “Blood vitamin D3 (25-OH-D3) levels are also correlated with the onset of COVID-19.” has been changed to “(line-171) In addition, the onset of COVID-19 is associated with low blood 25-OH-D3 levels.”.

“(line-239) 3.3 Vitamin D3 and chronic inflammation”

The new reference (3) has been added “(line-270) The consumption of fermented foods is associated with the numbers of equol-producing gut microbiota [50].”.

In “(line-919) Figure 1.” vitamin D3 and related terms were shown only in short names other than provitamin D3 (7-dehydrocholesterol) and abbreviations and alternative names have been described in figure legends. 

Summary of vitamin D related terms.

vitamin D2 (ergocalciferol)

provitamin D3 (7-dehydrocholesterol)

previtamin D3

vitamin D3 (cholecalciferol)

25-OH-D3 (calcidiol)

1α,25-(OH)2-D3 (calcitriol)

4(R),25-(OH)2-D3 (24,25-dihydroxycholecalciferol)

cholesterol

retinoid X receptor (RXR)

vitamin D receptor (VDR)

vitamin D binding proteins (VDBP)

vitamin D response element (VDRE)

cytochrome P450 2R1 (CYP2R1)

25-hydroxyvitamin D 1-α-hydroxylase (cytochrome p450 27B1: CYP27B1)

cytochrome P450 family 24 subfamily A member 1 (CYP24A1)

4. Still not all abbreviations are explained at first time use (see Abstract).

Response: All the abbreviations in the manuscript (including abstract) have been checked and added the meanings.

Abbreviations in Abstract (except vitamin D relevant terms)

tumor necrosis factor-α (TNF-α)

coronavirus disease 2019 (COVID-19)

Abbreviations in Body (except previously shown terms)

parathyroid hormone (PTH)

fibroblast growth factor 23 (FGF23)

ME-BYO (a non-disease condition)

recommended dietary allowance (RDA)

European Food Safety Authority (EFSA)

cathelicidin (LL-37)

lipopolysaccharide (LPS)

interferon-γ (IFN-γ)

interleukin-6 (IL-6)

toll-like receptor 2 (TLR2)

natural killer (NK)

interferon (IFN)

toll-like receptor 4 (TLR4)

mitogen-activated protein kinase kinase kinase 3 (MAP3Ks)

non-alcoholic fatty liver disease (NAFLD)

single-nucleotide polymorphism (SNP)

senescence-associated secretory phenotype (SASP)

non-alcoholic steatohepatitis (NASH)

inflammatory bowel disease (IBD)

cholic acid (CA)

chenodeoxycholic acid (CDCA)

farnesoid X receptor (FXR)

Takeda G protein-coupled receptor 5 (TGR5)

lithocholic acid (LCA)

Fermented grape food from Koshu (K-FGF)

drug delivery system (DDS)

nuclear factor-κ B (NF-κB)

glucagon-like peptide-1 (GLP-1)

interleukin-12 (IL-12)

interleukin-1β (IL-1β)

National Agriculture and Food Research Organisation (NARO)

proliferator-activated receptor γ coactivator 1-α (PGC1α).

AMP-activated protein kinase (AMPK)

nicotinamide mononucleotide (NMN)

non-alcoholic fatty liver disease (NFALD)

chronic obstructive pulmonary disease (COPD)

advanced glycation end-products (AGEs)

quality of life (QOL)

glycaemic-index (GI)

low-density lipoprotein (LDL)

triglyceride (TG)

glucan synthase-like1 (GSL1)

eicosapentaenoic acid (EPA)

docosahexaenoic acid (DHA)

Other changes

Section “(line-326) 4. Phytochemicals” has been revised. “4.1. Flavonoid polyphenols” has been deleted. In accordance with this change “4.2” has been changed to “(line-348) 4.1. Terpenoids” and “4.3” has been changed to “(line-360) 4.2. Carotenoids”. In addition, new subsection “(line-365) 4.3. Flavonoids ” has been added. The new reference (4) has been added “(line-358) [81]”. Furthermore, the reference “(line-424) [100]” has been changed to more suited reference. “(line-523) [130,131]” have been changed since the passages in the body have been updated in the writing process.

“(line-326) 4. Phytochemicals”

The passage “Especially, current review mainly focusses on phytochemicals contained in grapes that are particularly noteworthy.” has been changed to “(line-332) Polyphenols in grapes were one of the most studied phytochemicals.”.

The passage “The attractiveness of grape phytochemicals started with the red wine polyphony resveratrol, which gained the attention due to the “French Paradox” — where the mortality rate from cardiovascular disease is one third of the U.S.” has been changed to “(line-333) The attention to the grape phytochemicals started with the red wine polyphenol resveratrol, which gained the attention due to the “French Paradox” — the phenomenon of mortality rate from cardiovascular disease in French is one third of Americans.”

The passage “In addition, since the Mediterranean diet that is considered as healthy around the world contains lots of phytochemicals, they became famous [65].” has been changed to “(line-336) In addition, the Mediterranean diet, which is considered as healthy around the world contains lots of phytochemicals [68].”.

The subsection “4.1. Flavonoid polyphenols” has been deleted.

The subsection “4.2 Terpenoids” has been changed to “(line-348) 4.1. Terpenoids”.

The new reference (4) has been added “(line-358) Ursolic acid, another triterpene chemical, is also effective in skeletal muscle health, along with vitamin D3 [81].”.

The subsection “4.3 Carotenoids” has been changed to“(line-365) 4.2. Carotenoids”.

“(line-361) Well-known carotenoids include β-carotene, astaxanthin, β-cryptoxanthin, etc. The main carotenoids contained in grapes are β-carotene and lutein. β-Carotene is a vitamin A precursor mainly obtained from food intake; hence, grape phytochemicals are also helpful for the maintenance of vision [62,82,83].”.  

Between these passages, the new subsection was added “(line-365) 4.3. Flavonoids ”.

“(line-366) The flavonoids contained in grapes include catechins, flavan-3-ols, flavon-3-ols, and so on. The most abundant procyanidins in grapes are oligomeric procyanidins, the complexes of (epi)catechins. Furthermore, quercetin, a flavone-3-ol, is the second most abundant phytochemical in grapes after catechins, and it also exists in conjugates like quercetin-glycoside and quercetin-glucuronide [84,85,86]. Table 1 summarises the effects of flavonoids shown in this review.”

“(line-415) 5.1 Vitamin D3 and phytochemicals in bone metabolism”

The reference “[100]” in this passage was changed to more appropriate one ”(line-424) In addition, oleanolic acid suppresses inflammation by suppressing the production of interleukin-12 (IL-12) and TNF-α in dendritic cells [100]; cytokine production in Kupffer cells; and production of proinflammatory cytokines TNF-α, interleukin-1β (IL-1β), and IL-6 in macrophages [101].”.

“(line-479) 5.3 Vitamin D3 and phytochemicals in the suppression of chronic inflammation”

These references have been changed “(line-523) It is well known that vitamin D3 participate in the prevention of osteoporosis [130,131].”.

References added

27. Tong X.; Zhang Y.; Zhao Y.; Li Y.; Li T.; Zou H.; Yuan Y.; Bian J.; Liu Z.; Gu J. Vitamin D Alleviates Cadmium-Induced Inhibition of Chicken Bone Marrow Stromal Cells' Osteogenic Differentiation In Vitro. Animals (Basel). 2023, 13, 2544.

29. Rafique A.; Rejnmark L.; Heickendorff L.; Møller HJ. 25(OH)D3 and 1.25(OH)2D3 inhibits TNF-α expression in human monocyte derived macrophages. PLoS One 2019, 14, e0215383.

50. Sekikawa A.; Wharton W.; Butts B.; Veliky CV.; Garfein J.; Li J.; Goon S.; Fort A.; Li M.; Hughes TM. Potential Protective Mechanisms of S-equol, a Metabolite of Soy Isoflavone by the Gut Microbiome, on Cognitive Decline and Dementia. Int. J. Mol. Sci. 2022, 23, 11921.

81. Sakuma K.; Hamada K.; Yamaguchi A.; Aoi W. Current Nutritional and Pharmacological Approaches for Attenuating Sarcopenia. Cells 2023, 12, 2422.

The reference has been updated to more fit one.

100. Wei HJ.; Pareek TK.; Liu Q.; Letterio JJ. A unique tolerizing dendritic cell phenotype induced by the synthetic triterpenoid CDDO-DFPA (RTA-408) is protective against EAE. Sci. Rep. 2017, 7, 9886.

These references have been changed.

130. Maria S.; Swanson MH.; Enderby LT.; D'Amico F.; Enderby B.; Samsonraj RM.; Dudakovic A.; van Wijnen AJ.; Witt-Enderby PA. Melatonin-micronutrients Osteopenia Treatment Study (MOTS): a translational study assessing melatonin, strontium (citrate), vitamin D3 and vitamin K2 (MK7) on bone density, bone marker turnover and health related quality of life in postmenopausal osteopenic women following a one-year double-blind RCT and on osteoblast-osteoclast co-cultures. Aging (Albany NY). 2017, 9, 256-285.

131. Nakamura Y.; Suzuki T.; Kamimura M.; Ikegami S.; Uchiyama S.; Kato H. Alfacalcidol Increases the Therapeutic Efficacy of Ibandronate on Bone Mineral Density in Japanese Women with Primary Osteoporosis. Tohoku J. Exp. Med. 2017, 241, 319-326.

Reviewer 3 Report

Comments and Suggestions for Authors

The manuscript improved, but is is still not perfect:

1. The resolution of the figures is very insufficient, please improve.

2. Please have the nomenclature of genes also correct in figures (italic and latest nomenclature)

3. Please have a clear understanding of the vitamin D nomenclature and revise.

4. Still not all abbreviations are explained at first time use (see Abstract).

Comments on the Quality of English Language

Moderate corrections necessary.

Author Response

Comments and Suggestions for Authors

The manuscript improved, but is is still not perfect:

Thank you very much for your detailed review. In accordance with the point out by the reviewers we intensively checked our manuscript.

1. The resolution of the figures is very insufficient, please improve.

Response: Figures in the manuscript have been exported to PDF format from software used (Keynote Mac) and sent directly to the Editorial Office. We did not use Adobe Illustrator but figures in PDF format should be no problems.

2. Please have the nomenclature of genes also correct in figures (italic and latest nomenclature)

Response: In “(line-919) Figure 1” and “(line-942) Figure 3”, the name of cytokine proteins (TNF-α, IL-1β, IL-6) have been changed from italic to normal font. Figures were changed to show these proteins exist outside the nucleus.

Genes in the manuscript have been shown in the italic font.

TNF

CYP27B1

Klotho (KL)

VDR

CD14

CAMP

β-defensin 2 (DEFB4A)

SIRT1

SIRT7

3. Please have a clear understanding of the vitamin D nomenclature and revise.

Response: As the explanation of vitamin D3 has not been precise, passages in “1. Introduction” and “3. Vitamin D3” has been exchanged. The process of the generation of active form of vitamin D3 from provitamin D3, and process of inactivation have been summarised in the first paragraph of “3. vitamin D3”.

“(line-39) 1. Introduction”

The passage “The vitamin D3 precursor provitamin D3 (7-dehydrocholesterol) is converted to previtamin D3 and then to vitamin D3.” has been changed to “(line-55) Vitamin D3 is produced from cholesterol in skin cells after exposure to ultraviolet rays from sunlight.”.

“(line-109) 3. Vitamin D3”

The passage “Vitamin D3 is produced from cholesterol in skin cells after exposure to ultraviolet rays from sunlight.” has been changed to “(line-114) The vitamin D3 precursor provitamin D3 (7-dehydrocholesterol) is converted to previtamin D3 and then to vitamin D3.”.

The passage “Vitamin D3 circulates in the blood stream after binding to vitamin D-binding protein (VDBP) and is converted to 25-OH-D3 (calcidiol) by a liver enzyme.” has been changed to “(line-115) Vitamin D3 circulates in the blood stream after binding to vitamin D binding protein (VDBP) and is converted to 25-OH-D3 (calcidiol) by liver enzymes such as cytochrome P450 2R1 (CYP2R1).”.

The passage “Enzymes in the kidneys and immune cells convert 25-OH-D3 to the active form, 1α,25-(OH)2-D3 (calcitriol), which binds to the vitamin D receptor (VDR) in the cytoplasm and translocate into the nucleus [12,13,14].” has been change to “(line-117) An enzyme CYP27B1 in the kidneys and immune cells convert 25-OH-D3 to the active form, 1α,25-(OH)2-D3 (calcitriol), which binds to the vitamin D receptor (VDR) in the cytoplasm and translocate into the nucleus [12,13,14].”.

The passage has been added “(line-125) Activated form of vitamin D3 (1α,25-(OH)2-D3) inactivated to 24(R),25-(OH)2-D3 (24,25-dihydroxycholecalciferol) within a few hours by the enzyme CYP24A1.”.

The passage with new reference (1) has been added “(line-157) Vitamin D3 also mitigates the risk of bone density loss induced by cadmium poisoning [27].”.

“(line-159) 3.1 Vitamin D3 and infectious diseases ”

The new reference (2) has been added “(line-166) In this context, vitamin D3 stops the production of TNF-α and suppresses its responses to avoid inducing excess and chronic inflammation [29].”.

The passage “The onset of seasonal influenza and COVID-19 is associated with serum vitamin D3 (25-OH-D3) levels.” has been changed to “(line-169) The onset of seasonal flu is associated with serum 25-OH-D3 levels.”.

The passage “Blood vitamin D3 (25-OH-D3) levels are also correlated with the onset of COVID-19.” has been changed to “(line-171) In addition, the onset of COVID-19 is associated with low blood 25-OH-D3 levels.”.

“(line-239) 3.3 Vitamin D3 and chronic inflammation”

The new reference (3) has been added “(line-270) The consumption of fermented foods is associated with the numbers of equol-producing gut microbiota [50].”.

In “(line-919) Figure 1.” vitamin D3 and related terms were shown only in short names other than provitamin D3 (7-dehydrocholesterol) and abbreviations and alternative names have been described in figure legends. 

Summary of vitamin D related terms.

vitamin D2 (ergocalciferol)

provitamin D3 (7-dehydrocholesterol)

previtamin D3

vitamin D3 (cholecalciferol)

25-OH-D3 (calcidiol)

1α,25-(OH)2-D3 (calcitriol)

4(R),25-(OH)2-D3 (24,25-dihydroxycholecalciferol)

cholesterol

retinoid X receptor (RXR)

vitamin D receptor (VDR)

vitamin D binding proteins (VDBP)

vitamin D response element (VDRE)

cytochrome P450 2R1 (CYP2R1)

25-hydroxyvitamin D 1-α-hydroxylase (cytochrome p450 27B1: CYP27B1)

cytochrome P450 family 24 subfamily A member 1 (CYP24A1)

4. Still not all abbreviations are explained at first time use (see Abstract).

Response: All the abbreviations in the manuscript (including abstract) have been checked and added the meanings.

Abbreviations in Abstract (except vitamin D relevant terms)

tumor necrosis factor-α (TNF-α)

coronavirus disease 2019 (COVID-19)

Abbreviations in Body (except previously shown terms)

parathyroid hormone (PTH)

fibroblast growth factor 23 (FGF23)

ME-BYO (a non-disease condition)

recommended dietary allowance (RDA)

European Food Safety Authority (EFSA)

cathelicidin (LL-37)

lipopolysaccharide (LPS)

interferon-γ (IFN-γ)

interleukin-6 (IL-6)

toll-like receptor 2 (TLR2)

natural killer (NK)

interferon (IFN)

toll-like receptor 4 (TLR4)

mitogen-activated protein kinase kinase kinase 3 (MAP3Ks)

non-alcoholic fatty liver disease (NAFLD)

single-nucleotide polymorphism (SNP)

senescence-associated secretory phenotype (SASP)

non-alcoholic steatohepatitis (NASH)

inflammatory bowel disease (IBD)

cholic acid (CA)

chenodeoxycholic acid (CDCA)

farnesoid X receptor (FXR)

Takeda G protein-coupled receptor 5 (TGR5)

lithocholic acid (LCA)

Fermented grape food from Koshu (K-FGF)

drug delivery system (DDS)

nuclear factor-κ B (NF-κB)

glucagon-like peptide-1 (GLP-1)

interleukin-12 (IL-12)

interleukin-1β (IL-1β)

National Agriculture and Food Research Organisation (NARO)

proliferator-activated receptor γ coactivator 1-α (PGC1α).

AMP-activated protein kinase (AMPK)

nicotinamide mononucleotide (NMN)

non-alcoholic fatty liver disease (NFALD)

chronic obstructive pulmonary disease (COPD)

advanced glycation end-products (AGEs)

quality of life (QOL)

glycaemic-index (GI)

low-density lipoprotein (LDL)

triglyceride (TG)

glucan synthase-like1 (GSL1)

eicosapentaenoic acid (EPA)

docosahexaenoic acid (DHA)

Other changes

Section “(line-326) 4. Phytochemicals” has been revised. “4.1. Flavonoid polyphenols” has been deleted. In accordance with this change “4.2” has been changed to “(line-348) 4.1. Terpenoids” and “4.3” has been changed to “(line-360) 4.2. Carotenoids”. In addition, new subsection “(line-365) 4.3. Flavonoids ” has been added. The new reference (4) has been added “(line-358) [81]”. Furthermore, the reference “(line-424) [100]” has been changed to more suited reference. “(line-523) [130,131]” have been changed since the passages in the body have been updated in the writing process.

“(line-326) 4. Phytochemicals”

The passage “Especially, current review mainly focusses on phytochemicals contained in grapes that are particularly noteworthy.” has been changed to “(line-332) Polyphenols in grapes were one of the most studied phytochemicals.”.

The passage “The attractiveness of grape phytochemicals started with the red wine polyphony resveratrol, which gained the attention due to the “French Paradox” — where the mortality rate from cardiovascular disease is one third of the U.S.” has been changed to “(line-333) The attention to the grape phytochemicals started with the red wine polyphenol resveratrol, which gained the attention due to the “French Paradox” — the phenomenon of mortality rate from cardiovascular disease in French is one third of Americans.”

The passage “In addition, since the Mediterranean diet that is considered as healthy around the world contains lots of phytochemicals, they became famous [65].” has been changed to “(line-336) In addition, the Mediterranean diet, which is considered as healthy around the world contains lots of phytochemicals [68].”.

The subsection “4.1. Flavonoid polyphenols” has been deleted.

The subsection “4.2 Terpenoids” has been changed to “(line-348) 4.1. Terpenoids”.

The new reference (4) has been added “(line-358) Ursolic acid, another triterpene chemical, is also effective in skeletal muscle health, along with vitamin D3 [81].”.

The subsection “4.3 Carotenoids” has been changed to“(line-365) 4.2. Carotenoids”.

“(line-361) Well-known carotenoids include β-carotene, astaxanthin, β-cryptoxanthin, etc. The main carotenoids contained in grapes are β-carotene and lutein. β-Carotene is a vitamin A precursor mainly obtained from food intake; hence, grape phytochemicals are also helpful for the maintenance of vision [62,82,83].”.  

Between these passages, the new subsection was added “(line-365) 4.3. Flavonoids ”.

“(line-366) The flavonoids contained in grapes include catechins, flavan-3-ols, flavon-3-ols, and so on. The most abundant procyanidins in grapes are oligomeric procyanidins, the complexes of (epi)catechins. Furthermore, quercetin, a flavone-3-ol, is the second most abundant phytochemical in grapes after catechins, and it also exists in conjugates like quercetin-glycoside and quercetin-glucuronide [84,85,86]. Table 1 summarises the effects of flavonoids shown in this review.”

“(line-415) 5.1 Vitamin D3 and phytochemicals in bone metabolism”

The reference “[100]” in this passage was changed to more appropriate one ”(line-424) In addition, oleanolic acid suppresses inflammation by suppressing the production of interleukin-12 (IL-12) and TNF-α in dendritic cells [100]; cytokine production in Kupffer cells; and production of proinflammatory cytokines TNF-α, interleukin-1β (IL-1β), and IL-6 in macrophages [101].”.

“(line-479) 5.3 Vitamin D3 and phytochemicals in the suppression of chronic inflammation”

These references have been changed “(line-523) It is well known that vitamin D3 participate in the prevention of osteoporosis [130,131].”.

References added

27. Tong X.; Zhang Y.; Zhao Y.; Li Y.; Li T.; Zou H.; Yuan Y.; Bian J.; Liu Z.; Gu J. Vitamin D Alleviates Cadmium-Induced Inhibition of Chicken Bone Marrow Stromal Cells' Osteogenic Differentiation In Vitro. Animals (Basel). 2023, 13, 2544.

29. Rafique A.; Rejnmark L.; Heickendorff L.; Møller HJ. 25(OH)D3 and 1.25(OH)2D3 inhibits TNF-α expression in human monocyte derived macrophages. PLoS One 2019, 14, e0215383.

50. Sekikawa A.; Wharton W.; Butts B.; Veliky CV.; Garfein J.; Li J.; Goon S.; Fort A.; Li M.; Hughes TM. Potential Protective Mechanisms of S-equol, a Metabolite of Soy Isoflavone by the Gut Microbiome, on Cognitive Decline and Dementia. Int. J. Mol. Sci. 2022, 23, 11921.

81. Sakuma K.; Hamada K.; Yamaguchi A.; Aoi W. Current Nutritional and Pharmacological Approaches for Attenuating Sarcopenia. Cells 2023, 12, 2422.

The reference has been updated to more fit one.

100. Wei HJ.; Pareek TK.; Liu Q.; Letterio JJ. A unique tolerizing dendritic cell phenotype induced by the synthetic triterpenoid CDDO-DFPA (RTA-408) is protective against EAE. Sci. Rep. 2017, 7, 9886.

These references have been changed.

130. Maria S.; Swanson MH.; Enderby LT.; D'Amico F.; Enderby B.; Samsonraj RM.; Dudakovic A.; van Wijnen AJ.; Witt-Enderby PA. Melatonin-micronutrients Osteopenia Treatment Study (MOTS): a translational study assessing melatonin, strontium (citrate), vitamin D3 and vitamin K2 (MK7) on bone density, bone marker turnover and health related quality of life in postmenopausal osteopenic women following a one-year double-blind RCT and on osteoblast-osteoclast co-cultures. Aging (Albany NY). 2017, 9, 256-285.

131. Nakamura Y.; Suzuki T.; Kamimura M.; Ikegami S.; Uchiyama S.; Kato H. Alfacalcidol Increases the Therapeutic Efficacy of Ibandronate on Bone Mineral Density in Japanese Women with Primary Osteoporosis. Tohoku J. Exp. Med. 2017, 241, 319-326.

Comments on the Quality of English Language

Moderate corrections necessary.

Response: As we mentioned above English in the manuscript has been intensively checked and changed. 

We sincerely hope the changed manuscript is suitable to be published in International Journal of Molecular Sciences.

Round 3

Reviewer 3 Report

Comments and Suggestions for Authors

1. Leaving the figures out of the manuscript is not any solution to the problem. Without figures I cannot judge about their improvement.

2. We are now in the 3. round of review but there are still mistakes in the manuscript. For example, vitamin D3 is produced based on 7-dehydrocholesterol and not from cholesterol. Please check again all text and facts carefully throughout the whole manuscript.

Comments on the Quality of English Language

Some edits necessary.

Author Response

RESPONSES TO REVIEWERS

Response to Reviewer 3 - (Round 3)

Comments and Suggestions for Authors

Thank you very much for your review. We really appreciate your kindness.

1. Leaving the figures out of the manuscript is not any solution to the problem. Without figures I cannot judge about their improvement.

Response: We feel sorry to hear that figures were not in the manuscript. In the previously published manuscript, the Editorial Office kindly edited our manuscript and embedded figures to the word template hence we send the figures as same. However something different seemed to happen in this occasion.

In this manuscript, figures and table are located in the last part of the manuscript. We also send PDF figures separately to the Editorial Office, too. Final processes should be conducted by the Editorial Office. However, PDF version of the manuscript we submitted including vector drawing figures and embedded fonts.

2. We are now in the 3. round of review but there are still mistakes in the manuscript. For example, vitamin D3 is produced based on 7-dehydrocholesterol and not from cholesterol. Please check again all text and facts carefully throughout the whole manuscript.

Response: We revised the manuscript. Below are the changes we made.

“(line-39) 1. Introduction”

The passage “cholesterol” has been changed to “(line-55) precursor of cholesterol, 7-dehydrocholesterol (provitamin D3)”.

The new reference (1) has been added to the passage ”(line-67) According to the international definition, the level of 30 ng/mL is considered sufficient, 29-20 ng/mL is insufficient, and 19 ng/mL or less is considered as a deficiency [7].”.

The new reference (2) has been added to the passage ”(line-74) The half-life of active vitamin D3 (1α,25-(OH)-2-D3) is only several hours, and it is converted to inactivated vitamin D3, 24(R),25-(OH)2-D3 (24,25-dihydroxycholecalciferol), by cytochrome P450 family 24 subfamily A member 1 (CYP24A1) enzyme [9]. ”.

“(line-110) 3. Vitamin D3”

The passage “Figure 1 shows the activation pathway from provitamin D3 to the activated form (1α,25-(OH)2-D3) and inhibit the production of TNF-α.” has been changed to “(line-128) Figure 1 shows the pathway of vitamin D3 synthesis, from provitamin D3 to the activated form (1α,25-(OH)2-D3) and inhibition of TNF-α production.”.

“(line-209) 3.2 Vitamin D3 and natural immunity”

The new reference (3) has been added to the passage ”(line-236) During viral infections, type-1 interferon (IFN), IFN-α and IFN-β exert antiviral activity during viral infections, and CAMP/LL-37 exerts antiviral activity as well [43]. ”.

“(line-240) 3.3 Vitamin D3 and chronic inflammation”

The new reference (4) has been added to the passage ”(line-267) Vitamin D3 deficiency reduces the absorption of calcium from the digestive tract, and the intake of vitamin D3 and vitamin K is important for this pathway [53]. ”.

“(line-274) 3.4 Vitamin D3, gut microbiota, and gut environment”

The passages “However, the intake of a high-fat diet in humans increased Bacteroidetes and decreased Firmicutes. These findings indicated the complete differences between mice and humans.” has been changed to “(line-282) However, the intake of a high-fat diet in humans indicated the complete differences between mice and humans.”.

The passage “In European and American experiments comparing diabetic and healthy patients, Roseburia intestinalis and Faecalibacterium prausnitzii in the phylum Firmicutes were decreased and Bacteroidetes and Proteobacteria were increased [52,53].” has been changed to “(line-283) In European and American experiments comparing diabetic and healthy subjects, Roseburia intestinalis and Faecalibacterium prausnitzii in the phylum Firmicutes were decreased in diabetic subjects [56] and Bacteroidetes and Proteobacteria were increased in normal subjects [57].”.

The passage “In addition, in a comparison of healthy and type-2 diabetic Japanese patients, different results were observed, with a decrease in Bacteroidetes and increases in Firmicutes and Actinobacteria in diabetic patients [54].” has been changed to “(line-286) In addition, in a comparison of healthy and type-2 diabetic Japanese subjects, different results were observed, with a decrease in Bacteroidetes and increases in Firmicutes and Actinobacteria in diabetic subjects [58].”.

The passage “It is important to recognise the differences in gut microbiota because of the differences of ethnicity and dietary habits.” has been changed to “(line-288) Therefore, it is important to recognise the differences of gut microbiota depending on the differences between ethnicity and dietary habits.” and moved to last of the former paragraph.

The passage “In addition, many reports have supported the relationship between vitamin D3 and gut microbiota.” has been changed to “(line-291) Many reports have supported the relationship between vitamin D3 and gut microbiota.” and became the first passage of the next paragraph.

The passage “Bile acids secreted into the digestive tract act on micelle lipids and make them susceptible to decomposition by lipase.” has been changed to “(line-307) Bile acids secreted into the digestive tract modify lipids into micelles and make them susceptible to decomposition by lipase.”.

The passage “Figure 2 summarises the cellular distribution of FXR and TGR5 expression in the body.” has been changed to “(line-319) Figure 2 summarises the cellular distribution of bile acid receptors FXR and TGR5 expressed in the body.”.

“(line-366) 4.3. Flavonoids”

The passage “Table 1 summarises the effects of flavonoids shown in this review.” has been changed to “(line-371) Table 1 summarises the variety and effects of polyphenol flavonoids shown in this review.”.

“(line-416) 5.1 Vitamin D3 and phytochemicals in bone metabolism”

The passage “” has been changed to “(line-430) The effects of phytoestrogen equol and osteoporosis associated effect of β-cryptoxanthin have already been mentioned above [68].”. to meet the context of the manuscript.

Below are newly added references (1-4)

7. PÅ‚udowski P.; Kos-KudÅ‚a B.; Walczak M.; Fal A.; ZozuliÅ„ska-ZióÅ‚kiewicz D.; Sieroszewski P.; Peregud-Pogorzelski J.; Lauterbach R.; Targowski T.; LewiÅ„ski A.; SpaczyÅ„ski R.; WielgoÅ› M.; Pinkas J.; Jackowska T.; Helwich E.; Mazur A.; RuchaÅ‚a M.; Zygmunt A.; Szalecki M.; Bossowski A.; Czech-Kowalska J.; Wójcik M.; Pyrżak B.; Å»mijewski MA.; Abramowicz P.; Konstantynowicz J.; Marcinowska-Suchowierska E.; Bleizgys A.; Karras SN.; Grant WB,.; Carlberg C.; Pilz S.; Holick MF.; Misiorowski W. Guidelines for Preventing and Treating Vitamin D Deficiency: A 2023 Update in Poland. Nutrients 2023, 15, 695.

9. Hsu S.; Zelnick LR.; Lin YS.; Best CM.; Kestenbaum BR.; Thummel KE.; Hoofnagle AN.; de Boer IH. Validation of the 24,25-dihydroxyvitamin D3 to 25-hydroxyvitamin D3 ratio as a biomarker of 25-hydroxyvitamin D3 clearance. J. Steroid. Biochem. Mol. Biol. 2022, 217, 106047.

43. Bhatt T.; Dam B.; Khedkar SU.; Lall S.; Pandey S.; Kataria S.; Ajnabi J.; Gulzar SE.; Dias PM.; Waskar M.; Raut J.; Sundaramurthy V.; Vemula PK.; Ghatlia N.; Majumdar A.; Jamora C. Niacinamide enhances cathelicidin mediated SARS-CoV-2 membrane disruption. Front. Immunol. 2023, 14, 1255478.

53. Capozzi A.; Scambia G.; Lello S. Calcium, vitamin D, vitamin K2, and magnesium supplementation and skeletal health. Maturitas 2020, 140, 55-63.

Comments on the Quality of English Language

Some edits necessary.

Response: In addition to the changes shown above there are some minor changes: marked-up copy with yellow highlighting of any changes in the new document is attached as a “Revised Manuscript with Track Changes (supplementary)” file. 

Our manuscript may not be perfect, however we sincerely hope the manuscript must be sufficient enough to be published in International Journal of Molecular Sciences.

Round 4

Reviewer 3 Report

Comments and Suggestions for Authors

OK now

Comments on the Quality of English Language

minor edits necessary